# HUG3D: Human Group-Aware 3D Reconstruction from a Single Image with Physical Interaction

## Abstract

Reconstructing textured 3D human models from a single image is fundamental for AR/VR and digital human applications. However, existing methods predominantly focus on single individuals and thus fail in multi-human scenes, where naive composition of individual reconstructions often leads to artifacts such as unrealistic overlaps, missing geometry in occluded regions, and distorted interactions. These limitations highlight the need for approaches that incorporates group-level context and interaction priors. We introduce HUG3D, a holistic method that explicitly models both group- and instance-level information. To mitigate perspective-induced geometric distortions, we first transform the input into a canonical orthographic space. Our primary component, Human Group-aware Multi-View Diffusion (HUG-MVD), then generates complete multi-view normals and images by jointly modeling individuals and their group context to resolve occlusions and proximity. Subsequently, the Human Group-Aware Geometric Reconstruction (HUG-GR) module optimizes the geometry by leveraging explicit, physics-based interaction priors to enforce physical plausibility and accurately model inter-human contact. Finally, the multi-view images are fused into a high-fidelity texture. Extensive experiments show that HUG3D significantly outperforms both single-human and existing multi-human methods, producing physically plausible, high-fidelity 3D reconstructions of interacting groups from a single image.

## 1 Introduction

Reconstructing detailed 3D human models from visual input (Ho et al., 2024; Zhang et al., 2024; Jiang et al., 2024; Zheng et al., 2021; Li et al., 2024b; Saito et al., 2019; 2020; Xiu et al., 2022; Kim et al., 2023) is a fundamental task in computer vision, supporting applications in augmented and virtual reality (AR/VR) (Ma et al., 2021; Orts-Escolano et al., 2016), digital humans, and social behavior understanding. While monocular 3D reconstruction from a single RGB image (Ho et al., 2024; Zhang et al., 2024; Li et al., 2024b) has made substantial progress, most existing methods are limited to isolated individuals in controlled environments. However, these approaches often fail to generalize to real-world scenes involving multiple interacting people, where occlusion, perspective distortion, and spatial entanglement introduce significant ambiguity and modeling challenges.

In particular, we identify three core challenges in monocular multi-human 3D reconstruction: **(1) Geometric complexity and perspective distortion.** Multi-human scenes introduce strong perspective distortion due to depth variation, occlusion, and complex spatial layout. While most methods assume orthographic views leading to noticeable distortions on real-world perspective inputs (Fig. 1(a)), a few learn perspective-aware projections (Li et al., 2024a; Wang et al., 2025), but these are limited to single-object cases and struggle with multi-human complexity. The scarcity of annotated multi-human data (Yin et al., 2023; Zheng et al., 2021) further hinders generalization across camera poses and interaction patterns. **(2) Lack of interaction-aware geometric modeling.** Most methods reconstruct individuals independently, overlooking contextual cues like contact, occlusion, and spatial proximity (Fig. 1(b)). This often results in unrealistic outputs such as overlapping limbs or unnatural distances. Although group-wise SMPL-X approaches (Baradel et al., 2024; Müller et al., 2024; Sun et al., 2022) offer early signs of interaction modeling, full-surface, textured reconstruction remains underexplored. **(3) Missing geometry and texture in occluded regions.** Occlusions between

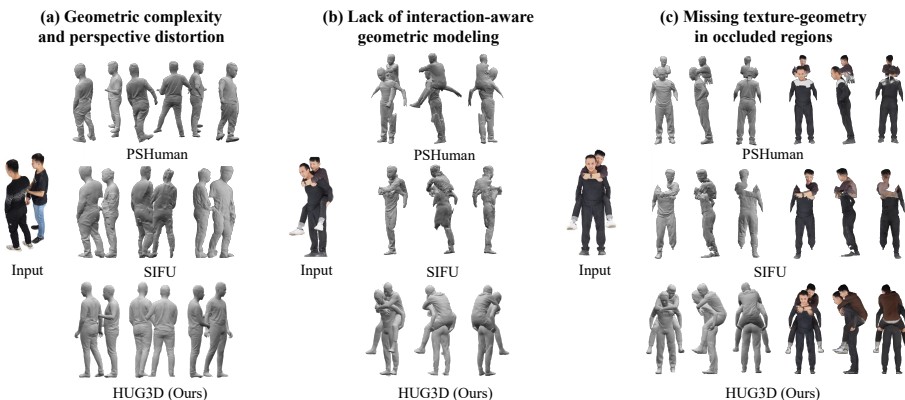

Figure 1: Core challenges in monocular multi-human 3D reconstruction: (a) geometric complexity and perspective distortion, (b) lack of interaction-aware geometric modeling, and (c) missing texture and geometry in occluded regions. Our HUG3D, jointly addresses all three challenges.

people obscure critical body parts, causing incomplete geometry and textures (Fig. 1(c)). While generative inpainting (Rombach et al., 2022; Labs, 2024) can hallucinate plausible content, only a few approaches (Cao et al., 2024; Barda et al., 2024) jointly address geometry and appearance, and even fewer try handling multi-view consistency under occlusion.

To address these challenges, we propose **HUG3D**, a holistic method for **HU**man **G**roup-aware **3D** reconstruction from a single image. HUG3D effectively incorporates both group- and instance-level information with three main components: (1) *Canonical Perspective-to-Orthographic View Transform (Pers2Ortho).* To make multi-view diffusion of interacting people tractable under severe geometric distortion, we transform the input perspective image into a canonical orthographic space. From the image, we estimate a partial 3D textured geometry and reproject it to obtain consistent multi-view RGB and normal representations. (2) *Human Group-Instance Multi-View Diffusion (HUG-MVD).* This diffusion model jointly completes missing geometry and textures while serving as an implicit prior that enforces physically plausible human interactions. (3) *Textured Mesh Reconstruction.* Our *Human Group-Instance Geometry Reconstruction (HUG-GR)* module first optimizes the mesh using group-level, instance-level, and physics-based supervision. Subsequently, the multi-view images are fused into a final, high-fidelity texture using occlusion-aware blending.

Experiments show that HUG3D outperforms single- and multi-human baselines, producing physically plausible, high-fidelity textured reconstructions 3D of interacting groups from a single image.

## 2 RELATED WORK

**Single-Human 3D Reconstruction.** Significant advances in single-human 3D reconstruction have been driven by parametric models such as SMPL (Loper et al., 2015) and SMPL-X (Pavlakos et al., 2019). Early methods fit models to 2D cues (Bogo et al., 2016), while later works regress parameters directly from images using deep networks (Kanazawa et al., 2018). Implicit and volumetric approaches (Saito et al., 2019; 2020; Mustafa et al., 2021; Fieraru et al., 2021) further improve geometric detail. Recent efforts enhance view consistency and texture. SIFU (Zhang et al., 2024) optimizes UVs via SDF, and SiTH (Ho et al., 2024) uses back-view synthesis. PSHuman (Li et al., 2024b) generates multi-view RGBs via diffusion, and LHM (Qiu et al., 2025) uses transformers over image and SMPL-X tokens. Though effective for isolated individuals, these methods struggle in multi-human settings. Naïve application per instance leads to artifacts like overlapping meshes and inconsistent scale, underscoring the need for interaction-aware models.

**Multi-Human 3D Reconstruction.** Reconstructing multiple humans in 3D remains challenging due to occlusions, inter-person interactions, and depth ambiguities. Early approaches reconstructed each individual independently (Li et al., 2019; Sun et al., 2022; Zheng et al., 2021), but this often led to physically implausible results. Later methods improved spatial coherence by introducing global constraints (Hassan et al., 2021) or jointly regressing shapes and poses in a shared coordinate system (Mustafa et al., 2021). Multi-view and video-based approaches (Kocabas et al., 2020; Jiang et al., 2024) further enhanced reconstruction accuracy, though they require video or multi-view inputs,

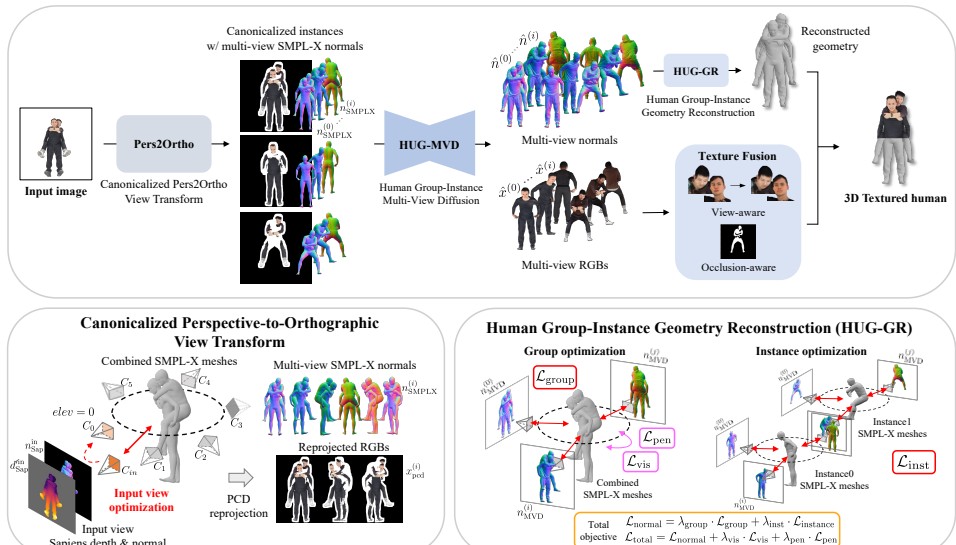

Figure 2: Overview of our HUG3D framework. Given a single perspective image, (1) the *Canonical Perspective-to-Orthographic View Transform (Pers2Ortho)* converts it into a consistent multi-view orthographic representation. (2) The *Human Group-Instance Multi-View Diffusion (HUG-MVD)* model completes occluded geometry and texture while maintaining plausible group interactions. (3) The *Textured Mesh Reconstruction* stage refines the mesh with our physics-aware *Human Group-Instance Geometry Reconstruction (HUG-GR)* module and generates high-fidelity textures.

which can be costly to acquire and process. More recently, learning-based methods have sought to tackle multi-human reconstruction from a single image. While interaction-aware networks (Fieraru et al., 2021; Cha et al., 2024; Chan et al., 2024) and model-free architectures (Mustafa et al., 2021) improve plausibility, their performance remains limited in single-view, single-frame scenarios.

**Scene-Level Human Reconstruction and Interaction Modeling.** Several methods incorporate human-scene or human-human interaction to improve physical plausibility. POSA (Hassan et al., 2021) enforces realistic body-ground contact but focuses on single-person scenarios. Group-level priors (Müller et al., 2024) enhance pose coherence, while BUDDI (Müller et al., 2024) learns a diffusion-based prior for plausible two-person interactions. Other approaches rely on explicit contact labels (Joo et al., 2018), but typically focus on coarse geometry or pose estimation. However, these methods often yield results with low texture fidelity or limited to SMPL-X mesh predictions, falling short of producing fully textured, detailed 3D reconstructions of interacting human groups.

# 3 HUMAN GROUP-AWARE 3D RECONSTRUCTION FROM A SINGLE IMAGE

Our HUG3D framework operates in three main stages, as illustrated in Fig. 2. First, the Canonical Perspective-to-Orthographic View Transform (*Pers2Ortho*) module converts the input perspective image into a consistent multi-view orthographic representation. Next, our Human Group-Instance Multi-View Diffusion (*HUG-MVD*) model completes occluded geometry and texture while ensuring plausible group interactions. Finally, the Textured Mesh Reconstruction stage refines the mesh with our physics-aware *HUG-GR* module and synthesizes a high-fidelity texture.

## 3.1 CANONICAL PERSPECTIVE-TO-ORTHOGRAPHIC VIEW TRANSFORM (PERS2ORTHO)

Multi-human scenes exhibit severe depth variation and occlusion, which violate the orthographic assumptions typically adopted in multi-view diffusion models. As a result, directly learning group interactions from a single perspective-view image is challenging due to extreme geometric complexity (see Fig. 3). To overcome this limitation, we introduce

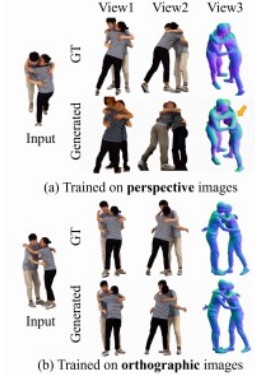

Figure 3: Comparison of results from multi-view diffusion trained on perspective vs. orthographic images.

Pers2Ortho module, a canonical view transformation module that estimates partial 3D geometry from the input perspective view and constructs consistent multi-view representations robust to geometric distortion.

**Initial Geometry, Segmentation, and Camera Setup.** Given a single RGB input, we first estimate an initial SMPL-X mesh, instance segmentation masks, and the perspective camera parameters $(\mathbf{K}_{\text{in}}, \mathbf{R}_{\text{in}}, \mathbf{T}_{\text{in}})$ of $\mathcal{C}_{\text{in}}$. We employ established methods based on SAM (Kirillov et al., 2023) and BUDDI (Müller et al., 2024), as detailed in Sec. A.1. To define a canonical processing space, the SMPL-X mesh is first normalized to a tight bounding box. Around this normalized geometry, we place six orthographic cameras $\{\mathcal{C}_0, \ldots, \mathcal{C}_5\}$ at fixed azimuths $\{0°, 45°, 90°, 180°, 270°, 315°\}$ with zero elevation. Each camera has extrinsic parameters $(\mathbf{R}_i, \mathbf{T}_i)$. This canonical camera rig ensures spatial alignment across all instances, providing a stable basis for downstream multi-view generation.

**Partial 3D Construction.** To enhance the fidelity of the canonical representation, we initialize a partial 3D mesh $\mathcal{M}$ with the initial SMPL-X mesh and refine it using geometric supervision from Sapiens (Khirodkar et al., 2024). Specifically, Sapiens predicts affine-invariant depth ($d_{\text{Sap}}^{\prime\text{in}}$) and surface normals ($n_{\text{Sap}}^{\text{in}}$) from the perspective input view $\mathcal{C}_{\text{in}}$. The mesh vertices of are optimized to align with these predictions by minimizing the geometry loss:

$$\mathcal{L}_{\text{geo}} = \left\| d_{\text{Sap}}^{\prime\text{in}} - d_{\mathcal{M}}^{\prime\text{in}} \right\|_2^2 + 1 - \left\langle n_{\text{Sap}}^{\text{in}}, n_{\mathcal{M}}^{\text{in}} \right\rangle, \tag{1}$$

where $d_{\mathcal{M}}^{\prime\text{in}}$ and $n_{\mathcal{M}}^{\text{in}}$ denote the depth and normal maps rendered from the mesh. The first term enforces depth consistency via L2 distance, while the second term enforces orientation consistency via cosine similarity. To handle possible topology mismatches between the initial mesh and Sapiens predictions, we adopt a remeshing strategy (Li et al., 2024b).

**Multi-View Input Generation via PCD Reprojection.** For the multi-view diffusion stage, we first render a complete set of normal maps $\{n_{\text{SMPLX}}^{(i)}\}_{i=0}^5$ from all six canonical views $\{\mathcal{C}_i\}$ from the initial SMPL-X mesh for the pose guidance.

Second, we generate partial RGB inputs $\{x_{\text{pcd}}^{(0)}, x_{\text{pcd}}^{(1)}, x_{\text{pcd}}^{(5)}\}$ by reprojecting the input image onto the refined partial 3D mesh $\mathcal{M}$ from three key canonical views ($0°$, $45°$, and $315°$). Concretely, we render a depth map $d_{\mathcal{M}}^{\prime\text{in}}$ from $\mathcal{M}$ and build a dense point cloud (PCD) $\mathcal{P}$ in the coordinate system of $\mathcal{C}_{\text{in}}$. This PCD is reprojected into each orthographic view $\mathcal{C}_i$ as:

$$\mathcal{P}_i = \Pi_{\text{ortho}}^i(\mathcal{P}) = \mathbf{R}_i \cdot \mathcal{P} + \mathbf{T}_i, \tag{2}$$

where $\Pi_{\text{ortho}}^i$ denotes the orthographic projection. RGB values from $\mathcal{C}_{\text{in}}$ are then transferred onto $\mathcal{P}_i$, yielding partial RGB maps $x_{\text{pcd}}^{(i)}$. Unlike mesh vertex coloring, which often produces sparse and low-quality textures, our PCD reprojection preserves dense appearance details in visible regions while maintaining spatial consistency across canonical views. These inputs serve as robust conditioning signals for multi-view diffusion. Further details can be found in Sec. A.2 of the supplementary.

## 3.2 HUMAN GROUP-INSTANCE MULTI-VIEW DIFFUSION (HUG-MVD)

We introduce Human Group-Instance Multi-View Diffusion (HUG-MVD), a diffusion model that leverages both group-level and instance-level priors to resolve occlusions and inter-person interactions in multi-human scenes.

**Interaction- and Occlusion-Aware Multi-View Diffusion.** Given reprojected partial RGB inputs and canonical-view SMPL-X normal maps, HUG-MVD reconstructs geometry and appearance missing due to occlusion. Unlike standard single-human diffusion models, our formulation incorporates group-level priors by jointly training on two complementary sources. First, diverse single-human datasets with full supervision (Ho et al., 2023; Yu et al., 2021) provide coverage over a wide range of body shapes and identities. Second, multi-human datasets with partial geometry (Yin et al., 2023) capture realistic inter-person occlusions and interactions, though with limited identity variation. This combination enables the model to generalize across diverse individuals while remaining robust to complex group-level occlusions.

To simulate realistic occlusions for single-human training data, we generate masked inputs $x_{\text{mask}}^{(i)}$ from the reprojected point clouds $x_{\text{pcd}}^{(i)}$ and visibility masks $M_{\text{vis}}$, as described in Sec. A.3.3. These

masks indicate unobserved regions for each canonical view $\mathcal{C}_i$. The model is conditioned on SMPL-X normal maps $\{n_{\text{SMPLX}}^{(i)}\}_{i=0}^5$, which provide geometric guidance via ControlNet (Zhang et al., 2023), steering the denoising process toward consistent, human-like reconstructions.

**Multimodal Training Objective.** Given 6 canonical views, the model predicts complete RGB images $\{\hat{x}^{(i)}\}$ and surface normals $\{\hat{n}^{(i)}\}$ simultaneously using a shared denoising diffusion process. To train the model, we minimize the following objective:

$$\mathcal{L}_{\text{diff}} = \sum_{i=0}^5 \left( \mathbb{E}_{t,\epsilon} \left[ \left\| \epsilon - \epsilon_\theta(z_{t,\text{rgb}}^{(i)}, t, x_{\text{mask}}^{(i)}, n_{\text{SMPLX}}^{(i)}) \right\|_2^2 \right] + \mathbb{E}_{t,\epsilon} \left[ \left\| \epsilon - \epsilon_\theta(z_{t,\text{normal}}^{(i)}, t, x_{\text{mask}}^{(i)}, n_{\text{SMPLX}}^{(i)}) \right\|_2^2 \right] \right),$$
(3)

where $z_{t,\text{rgb}}^{(i)}$ and $z_{t,\text{normal}}^{(i)}$ are the noisy latents for the RGB and normal map at timestep $t$, respectively. The denoising network $\epsilon_\theta$ learns to estimate the clean signals for both modalities, conditioned on the masked RGB inputs $x_{\text{mask}}^{(i)}$ and SMPL-X normal maps $n_{\text{SMPLX}}^{(i)}$. By jointly optimizing RGB and normal prediction, the model enforces consistency between appearance and geometry.

**Joint Group-Instance Inference.** At inference, a single HUG-MVD model jointly performs group- and instance-level reconstruction, predicting complete normals $\{\hat{n}^{(i)}\}$ and RGB images $\{\hat{x}^{(i)}\}$ from partial RGB inputs and SMPL-X guidance.

To maintain consistency between group- and instance-level latent representations, instance-specific latents $z_{t,\text{inst}(k)}^{(i)}$ are injected into the group latent $z_{t,\text{group}}^{(i)}$ at their corresponding spatial regions. Formally, at diffusion timestep $t$:

$$z_{t,\text{group}}^{(i)} \leftarrow \sum_{k=1}^K \left[ \alpha \cdot \mathbb{1}_{\text{inst}}^{(i,k)} \cdot z_{t,\text{inst}(k)}^{(i)} + (1-\alpha) \cdot \mathbb{1}_{\text{inst}}^{(i,k)} \cdot z_{t,\text{group}}^{(i)} \right] + \left( 1 - \sum_{k=1}^K \mathbb{1}_{\text{inst}}^{(i,k)} \right) \cdot z_{t,\text{group}}^{(i)}, \quad (4)$$

where $\mathbb{1}_{\text{inst}}^{(i,k)}$ indicates the spatial extent of instance $k$ in view $\mathcal{C}_i$, and $\alpha \in [0,1]$ controls blending strength. This composition allows fine-grained instance details to inform the global group representation, improving geometric consistency across overlapping surfaces.

Further details with detailed illustrations of HUG-MVD are provided in Sec. A.3 of the supplementary.

## 3.3 TEXTURED MESH RECONSTRUCTION

### 3.3.1 HUMAN GROUP-INSTANCE GEOMETRY RECONSTRUCTION (HUG-GR)

We refine the initial SMPL-X mesh to produce physically plausible geometry for multi-human scenes, leveraging both group- and instance-level supervision and physics-inspired constraints.

**Group-Instance Normal Supervision.** The mesh $\mathcal{M}$ is optimized to match predicted normals from HUG-MVD at both group and instance levels. Let $n_{\text{MVD,group}}^{(i)}$ and $n_{\mathcal{M},\text{group}}^{(i)}$ denote predicted and rendered normals for the group, and $n_{\text{MVD,inst}}^{(i,k)}$ and $n_{\mathcal{M},\text{inst}}^{(i,k)}$ for instance $k$. We define:

$$\mathcal{L}_{\text{normal}} = \lambda_{\text{group}} \cdot \mathcal{L}_{\text{group}} + \lambda_{\text{inst}} \cdot \mathcal{L}_{\text{instance}}, \quad (5)$$

$$\mathcal{L}_{\text{group}} = \sum_{i=0}^5 \left( 1 - \langle n_{\text{MVD,group}}^{(i)}, n_{\mathcal{M},\text{group}}^{(i)} \rangle \right), \quad \mathcal{L}_{\text{instance}} = \sum_{i=0}^5 \sum_{k=1}^K \left( 1 - \langle n_{\text{MVD,inst}}^{(i,k)}, n_{\mathcal{M},\text{inst}}^{(i,k)} \rangle \right), \quad (6)$$

where $n_{\mathcal{M}}$ is rendered from the currently optimized mesh via a differentiable renderer at each step.

**Interpenetration Loss.** To prevent implausible overlaps between body parts, we define an interpenetration loss over tolerance pairs $(i,j) \in \Omega_{\text{tol}}$, derived from contact regions in the initial SMPL-X meshes. For each pair, let $s_1^{i,j}$ and $s_2^{i,j}$ be the closest points on the surfaces of parts $i$ and $j$. The loss penalizes distances below a threshold $tol$, discouraging overlaps.:

$$\mathcal{L}_{\text{pen}} = \frac{1}{|\Omega_{\text{tol}}|} \sum_{(i,j) \in \Omega_{\text{tol}}} \left( |s_1^{i,j} - s_2^{i,j}| + \max \left( 0, \ tol - |s_1^{i,j} - s_2^{i,j}| \right) \right). \quad (7)$$

**Visibility Loss.** We enforce consistency between rendered visibility and ground-truth masks:

$$\mathcal{L}_{\text{vis}} = \frac{1}{2B} \sum_{k=1}^K \sum_{b=1}^B \frac{E_b^k}{M_b^k + \epsilon}, \quad (8)$$

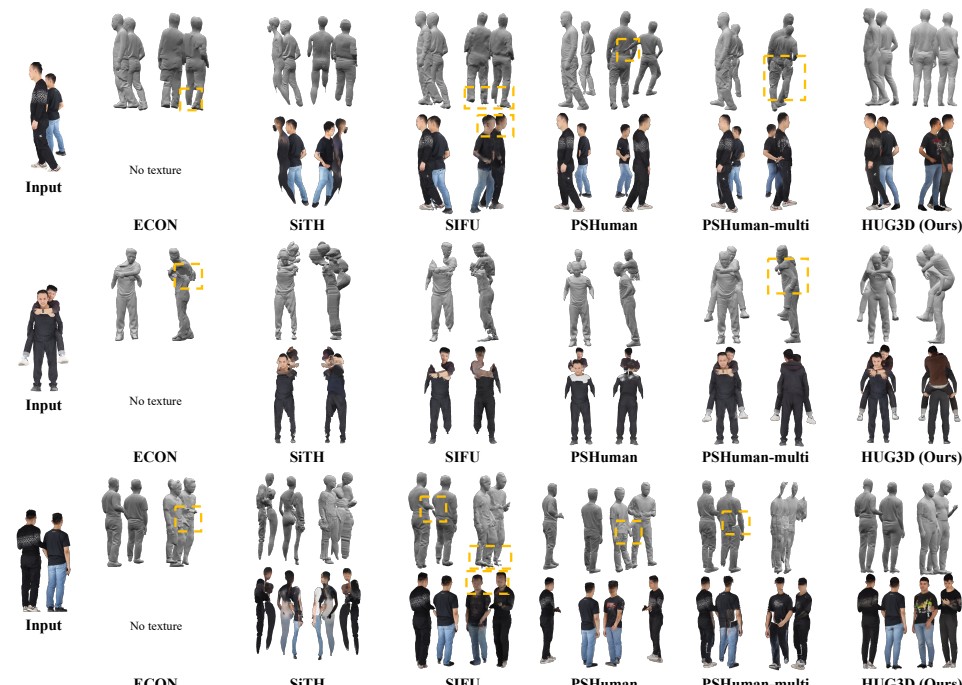

Figure 4: Qualitative comparison on multi-human 3D reconstruction from a single image. HUG3D outperforms baselines by correcting perspective distortion, preserving inter-human contact, and hallucinating plausible textures under heavy occlusion.

where $E_b^k$ and $M_b^k$ denote incorrectly occluded and total visible pixels for body part $b$ in instance $k$. This loss encourages each body part in the rendered mesh to match its expected visible region, ensuring accurate silhouettes and occlusion boundaries, especially in complex multi-human interactions.

**Total Optimization Objective.** The final mesh is optimized using:

$$\mathcal{L}_{\text{total}} = \mathcal{L}_{\text{normal}} + \lambda_{\text{vis}} \cdot \mathcal{L}_{\text{vis}} + \lambda_{\text{pen}} \cdot \mathcal{L}_{\text{pen}}. \tag{9}$$

We apply finer learning rates to high-frequency semantic regions (e.g., hands, face) to improve local accuracy while maintaining overall shape stability.

Additional details for HUG-GR are provided in Sec. A.4 of the supplementary.

### 3.3.2 TEXTURE CONSTRUCTION

Full-body vertex texture is generated by projecting multi-view RGBs onto the optimized mesh. To improve fidelity, occluded and low-confidence regions are blended using view-aware confidence masks. High-fidelity face restoration is applied for oblique or occluded views. Please refer to Sec. A.5 of the supplementary for additional details on texture construction.

## 4 EXPERIMENTS

### 4.1 EXPERIMENTAL SETUP

**Implementation Details.** The *Pers2Ortho* module is based on SAM (Kirillov et al., 2023) for instance mask extraction and BUDDI (Müller et al., 2024) for SMPL-X fitting and camera parameter estimation. For partial 3D construction, the initial SMPL-X mesh is optimized for 200 iterations using Adam (Kingma & Ba, 2014) with a learning rate of 0.02. *HUG-MVD* is initialized from PSHuman (Li et al., 2024b), based on Stable Diffusion 2.1 Rombach et al. (2022), and integrates a frozen normal-map ControlNet (Zhang et al., 2023). Training is performed on a single NVIDIA A100 (80GB) GPU with a batch size of 16 and gradient accumulation of 8 steps, using Adam (lr=$5 \times 10^{-6}$, $\beta_1 = 0.9$, $\beta_2 = 0.999$). A two-stage curriculum is employed: (1) 1,000 steps without inpainting masks, and (2) 1,000 steps with inpainting masks to simulate occlusion. Training takes approximately

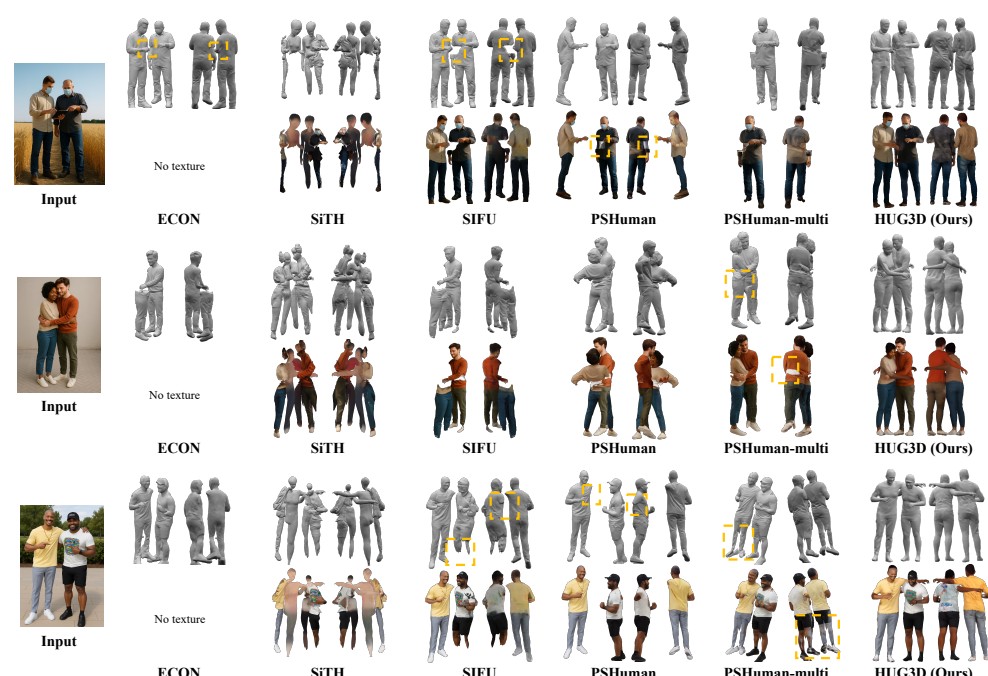

Figure 5: Qualitative results on in-the-wild images. Our method demonstrates robust multi-human reconstructions, outperforming baseline approaches and highlighting its practical applicability.

two days. A DDPM scheduler with 1,000 diffusion steps is used during training, while inference uses a DDIM scheduler ($\eta = 1.0$) with 40 denoising steps. A blending factor of $\alpha = 0.8$ balances diversity and fidelity. *HUG-GR* optimizes the mesh over 200 iterations with a learning rate of 0.01 and loss weights: $\lambda_{group} = 1.0$, $\lambda_{inst} = 0.2$, $\lambda_{pen} = 2.0$, and $\lambda_{vis} = 1.0$. Further implementation details are provided in Sec. A of the supplementary.

**Training Data.** Training samples of HUG-MVD are rendered from raw scans which are composed of textured mesh sequences. We train our model using the Hi4D dataset (Yin et al., 2023) for multi-human supervision, along with THuman2.0 (Yu et al., 2021) and CustomHumans (Ho et al., 2023) for diverse single-human poses and appearances, enabling robust learning across varied interactions and body configurations. Additional details are provided in Sec. A.3.1 of the supplementary.

**Evaluation Dataset.** We conduct experiments on the MultiHuman (Zheng et al., 2021) dataset, which provides multi-person, multi-view sequences with full 3D mesh supervision. For quantitative evaluation, we focused on two-person interacting scenes, including 20 sequences - six of these feature closely interacting pairs with heavy occlusions and complex spatial entanglements, while remainings capture more natural interactions. For each scene, we select a random initial viewpoint and render four perspective views, to evaluate under natural camera distortion, at azimuth offsets of $\{0°, 90°, 180°, 270°\}$, yielding 80 images in total. See Sec. B.2 of the supplementary for the details.

**Baselines.** To the best of our knowledge, we are the *first* to tackle *multi-human 3D reconstruction with both geometry and texture*, and no existing public baselines directly address this task (see Tab. S5). We therefore compare our method against two categories of prior works: (i) single-human reconstruction from a single image, and (ii) multi-human reconstruction from multi-view images or videos. To ensure a fair comparison, we follow the evaluation protocol of (Cha et al., 2024). For single-human methods—ECON (Xiu et al., 2023), SIFU (Zhang et al., 2024), SiTH (Ho et al., 2024), and PSHuman (Li et al., 2024b)—we crop each person using the dataset's ground-truth instance masks, reconstruct them independently, and then align the results into a shared coordinate frame and composited to form the complete scene. We also report PSHuman's performance when applied directly to uncropped multi-person images. For multi-human methods, we evaluate DeepMultiCap (Zheng et al., 2021), designed for multi-view images, and Multiply (Jiang et al., 2024), designed for videos, under single image setting. All SMPL-based methods use ground-truth SMPL-X poses to isolate

Table 1: Quantitative comparison of geometric metrics for multi-human 3D reconstruction. HUG3D achieves the best overall scores in all metrics including CD, P2S, and NC, and also outperforms other baselines in CP, indicating better interaction-aware reconstruction.

| Method | CD ↓ | P2S ↓ | NC ↑ | F-score ↑ | bbox-IoU ↑ | Norm $L2$ ↓ | CP ↑ |
|---|---|---|---|---|---|---|---|
| SIFU | 5.644 | 2.284 | 0.754 | 29.244 | 0.778 | 0.028 | 0.089 |
| SiTH | 9.251 | 3.185 | 0.709 | 21.037 | 0.708 | 0.040 | 0.135 |
| PSHuman | 15.579 | 6.088 | 0.617 | 9.749 | 0.659 | 0.069 | 0.027 |
| DeepMultiCap | 13.719 | 2.555 | 0.749 | 18.125 | 0.513 | 0.049 | 0.083 |
| **Ours** | **3.631** | **1.752** | **0.811** | **41.504** | **0.847** | **0.019** | **0.240** |

Table 2: Quantitative evaluation on texture quality of multi-human 3D reconstruction.

| Method | PSNR ↑ | SSIM ↑ | LPIPS ↓ |
|---|---|---|---|
| SIFU | 15.202 | 0.793 | 0.202 |
| SiTH | 13.798 | 0.789 | 0.233 |
| PSHuman | 11.418 | 0.742 | 0.304 |
| **Ours** | **16.456** | **0.809** | **0.168** |

Table 3: Quantitive evaluation within occluded region.

| Method | Norm $L2$ ↓ | PSNR ↑ | SSIM ↑ |
|---|---|---|---|
| SIFU | 0.197 | 6.157 | 0.559 |
| SiTH | 0.197 | 6.355 | 0.539 |
| PSHuman | 0.252 | 4.621 | 0.510 |
| DeepMultiCap | 0.217 | - | - |
| **Ours** | **0.140** | **8.388** | **0.602** |

Table 4: Ablation study of key components in HUG3D. We report geometry (CD, P2S, Norm $L2$) and texture (PSNR, LPIPS) metrics under various configurations.

| Module | Method | CD ↓ | P2S ↓ | Occ.Norm $L2$ ↓ | PSNR ↑ | LPIPS ↓ | Occ.PSNR ↑ |
|---|---|---|---|---|---|---|---|
| HUG-MVD | Trained on group-only data | 4.564 | 2.203 | 0.157 | 15.641 | 0.191 | 7.423 |
| | Trained on instance-only data | 4.645 | 2.245 | 0.156 | 15.840 | 0.185 | 7.726 |
| | w/o Instance-to-group latent composition | 4.646 | 2.249 | 0.159 | 16.198 | 0.183 | 7.916 |
| HUG-GR | Instance-only normal supervision | 4.642 | 2.250 | 0.156 | 16.180 | 0.183 | 7.902 |
| | Group-only normal supervision | 4.620 | 2.230 | 0.159 | 16.169 | 0.183 | 7.678 |
| | **Ours (Full)** | **4.316** | **2.122** | **0.153** | **16.454** | **0.179** | **8.082** |

reconstruction quality from pose estimation errors. Further details are provided in B.1 and results on multi-human baselines can be found in Secs. C.1 of the supplementary material.

**Evaluation Metrics.** We evaluate geometry with $L_1$ Chamfer distance (CD) [cm] and 1-directional point-to-surface distance (P2S) [cm], normal consistency (NC), F-score, and bbox-IoU between reconstructed and ground-truth meshes. To assess surface detail consistency, we compute $L_2$ normal error between predicted and ground-truth normal renders - across four rotated views at $\{0°, 90°, 180°, 270°\}$ relative to the input view. This normal error is separately computed specifically within occluded regions to evaluate robustness under occlusions. Physical realism is quantified through contact precision score (CP) defined by the overlap between the estimated and ground-truth inter-body contact map. Texture fidelity is assessed using PSNR, SSIM, and LPIPS, computed across four rotated views same as $L_2$ normal error. Also separately computed specifically within occluded regions. More details on the metrics are provided in Sec. B.3 of the supplementary.

## 4.2 RESULTS

**Quantitative Results.** As shown in Tab. 1, our method outperforms on all geometric metrics, with the lowest CD, P2S and highest NC. CP is markedly higher than other baselines, indicating superior physical realism and fidelity of inter-instance contacts. Tab. 2 shows our method achieved the highest PSNR and SSIM scores along with the lowest LPIPS. Moreover, in the occluded-region evaluation, Tab. 3, we significantly outperform the baselines in both Normal $L2$ error for geometry and PSNR/SSIM for texture. This indicates better perceptual quality of hallucinated textures where baselines typically fall.

**Qualitative Results.** Fig. 4 illustrates that our approach surpasses all baselines in multi-human 3D reconstruction from a single image. Pers2Ortho corrects perspective distortion, enabling accurate shape recovery in challenging viewpoints such as elevated scenes, while baseline models produce distorted results. Through HUG-MVD and HUG-GR, we model inter-human interactions, producing meshes that faithfully align with contact regions in the input image. In contrast, baselines either suffer from interpenetration or fail to preserve contact. Visual inspections confirm that HUG3D successfully reconstructs complete human meshes even under substantial occlusion. Hallucinated

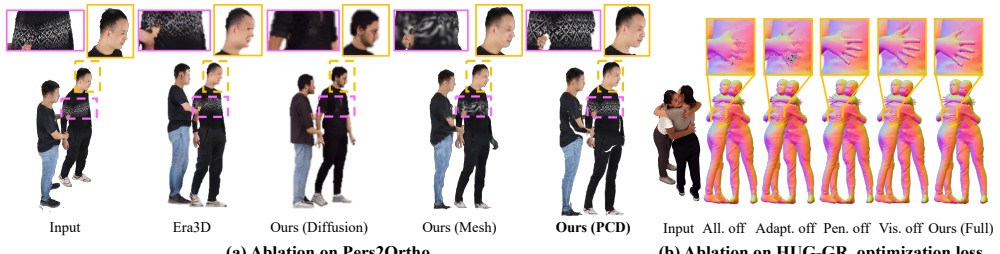

Figure 6: Ablation study on key components. (a) Pers2Ortho improves projection sharpness.(b) Each HUG-GR loss boosts geometric plausibility.

textures plausibly infer clothing and shape details not visible in the input. Competing methods, by contrast, exhibit broken surfaces, floating limbs, and missing textures. Fig. 5 shows results on in-the-wild images. Our approach consistently reconstructs multiple humans with high fidelity, surpassing baseline methods and demonstrating real-world applicability.

Additional results can be found in Secs. C and D of the supplementary.

### 4.3 ABLATION STUDY

**Pers2Ortho.** As shown in Fig. 6(a), we evaluate our PCD-based Pers2Ortho against several baselines: the generative method Era3D (Li et al., 2024a), and our custom diffusion and mesh reprojection variants. These competing methods often produce blurry results with lost facial details, whereas our strategy consistently preserves high-fidelity features from the original view.

**HUG-MVD.** Tab. 4 shows that training HUG-MVD on group-only data (Yin et al., 2023) suffers from limited geometry and texture diversity, while instance-only data (Ho et al., 2023; Yu et al., 2021) fails to capture inter-person interactions. Our full setting, combining both sources, significantly outperforms either alone, highlighting their complementary benefits. Also, disabling instance-to-group latent composition leads to inconsistent surfaces and breaks continuity across instances.

**HUG-GR.** We analyze the impact of each component in HUG-GR. As shown in Fig. 6(b), omitting adaptive-region specific optimization degrades quality in high-frequency areas, such as hands. Removing interpenetration loss leads to mesh interpenetration and unrealistic overlaps, demonstrating its importance for physically grounded reconstruction. Excluding the visibility loss impairs alignment and produces unnatural surface at contact regions. In Tab. 4, by comparing variants using only group-level or only instance-level losses, we found that incorporating both losses simultaneously yields the best performance on both geometry and texture quality.

More ablations and analysis can be found in Sec. E of the supplementary.

## 5 DISCUSSION

**Limitations.** HUG3D focuses on inter-human occlusion and does not handle occlusions from external objects, which we plan to address in future work. Although robust in the wild, our method depends on SMPL-X initialization and may fail under severe depth ambiguity or heavy occlusion, leading to distorted geometry. Further discussion appears in Sec. G of the supplementary.

**Conclusion.** We presented HUG3D, a three-stage framework for high-fidelity 3D reconstruction of human groups from a single RGB image. HUG3D addresses key challenges such as occlusions, complex interactions, and geometric distortions by combining a canonical view transform (Pers2Ortho), a group-aware multi-view diffusion model (HUG-MVD), and a physics-based reconstruction stage (HUG-GR) for refined geometry and texture. Extensive experiments show that HUG3D significantly outperforms single-human and prior multi-human methods, producing visually accurate, physically plausible reconstructions. Our framework enables reliable applications in AR/VR, telepresence, and digital human modeling, advancing realistic multi-human 3D reconstruction from a single image.

ETHICS STATEMENT

We take ethics very seriously, and our work fully conforms to the ICLR Code of Ethics. While our method enables legitimate applications such as AR/VR, telepresence, and graphics, it also carries potential dual-use risks, including privacy violations, lack of consent, biometric identification, and harassment. We rely exclusively on publicly available datasets released by third parties and did not collect any new human-subject data; therefore, IRB approval was not required. To mitigate misuse, we do not release pretrained weights. Any released code is intended for research purposes only and is provided under a license that prohibits non-consensual modeling, surveillance, re-identification, sexual or exploitative uses, and any use involving minors without verified consent. All demonstrations use either consented or synthetic inputs, and we enforce strict access controls with minimal data retention. We also provide detailed documentation of the limitations and typical failure cases of our method, and we clearly label all generated assets.

REPRODUCIBILITY STATEMENT

We follow ICLR guidance by providing a concise statement that points to the materials required to reproduce our results. The overall pipeline and experimental protocol are described in Sec. 3. Implementation and training details for each stage are provided in the supplementary material, including Secs. A.2, A.3, A.4, and A.5. Evaluation settings, datasets, and metrics are detailed in Secs. B.1, B.2, and B.3. Adaptations of baselines and additional comparisons appear in Sec. C.1, while efficiency and compute details are summarized in Sec. E.5. Information about environment libraries, datasets, and pretrained components used for research-only evaluation is listed in Secs. F.1, F.2, and F.3. Upon acceptance, we will release training and inference code along with reproduction scripts that rebuild all main tables and figures using only the public datasets cited in this paper. In line with our Ethics Statement, we will not release pretrained weights due to potential risks of non-consensual modeling and re-identification.

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

# Supplementary Material

L**IST OF** C**ONTENTS**

## A DETAILS ON METHODS AND IMPLEMENTATION

### A.1 ROBUST INSTANCE SEGMENTATION AND SMPL-X ESTIMATION

**Instance Segmentation.** We adopt a hybrid instance segmentation pipeline that combines detection, pose estimation, and promptable segmentation to produce per-person masks from a single image. This process is designed to be occlusion-aware and ensures that each human instance is segmented consistently, serving as the foundation for downstream SMPL-X fitting.

We begin by applying YOLOv11 (Khanam & Hussain, 2024) to detect human instances in the input image, using only the "person" class to extract tight bounding boxes around each individual. For each detected bounding box, we then estimate 2D keypoints using ViTPose (Xu et al., 2022) generated from BUDDI (Müller et al., 2024) project. These keypoints provide reliable localization of body joints and are retained for subsequent matching and alignment purposes. To generate high-quality binary masks for each person, we pass the bounding boxes as prompts to the Segment Anything Model (SAM) (Kirillov et al., 2023), which produces accurate per-instance segmentations. These masks are then associated with individual people by solving a Hungarian assignment problem between keypoint-based anchor regions (e.g., head and feet) and the detected masks, ensuring proper instance-level alignment. To further improve segmentation quality, overlapping or duplicate detections are merged based on intersection-over-union (IoU) thresholds. Additionally, in cases of occlusion or ambiguous limb segmentation, we use the predicted keypoints along with SAM's region prompting to correct mismatched or missing parts, particularly in the hand and foot regions. This pipeline enables robust and scalable segmentation of multiple humans in a single view, even in the presence of occlusion or complex poses.

**SMPL-X Estimation (RoBUDDI).** We adopt BUDDI (Müller et al., 2024), a diffusion-based prior model, to estimate SMPL-X parameters for multi-human scenes. While BUDDI produces high-quality predictions for individual subjects, it exhibits limited effectiveness in handling collisions and interpenetrations. This is primarily because the penetration constraints are applied only in a second-stage refinement, after collisions have already occurred. As a result, scenes with dense interactions still suffer from body interpenetrations and inaccurate keypoint estimations in occluded regions.

To overcome these limitations, we introduce two physics-inspired supervision terms during optimization: (1) an interpenetration loss that penalizes body collisions between interacting subjects, and (2) a visibility-aware keypoint loss that reduces errors in self- and inter-human occluded areas. These additions enable more robust and physically plausible multi-human pose estimation. We refer to our enhanced approach as RoBUDDI, which integrates these geometry-level constraints into the fitting process. As shown in Tab. S10 and Fig. S21, RoBUDDI achieves both quantitatively superior accuracy and qualitatively improved physical realism compared to the baseline.

The interpenetration loss penalizes unrealistic mesh overlaps between specific body part pairs. We define a tolerance set $\Omega_{\text{tol}}$, which enumerates body part index pairs that are likely to collide in crowded scenes (e.g., left thigh vs. right calf). For each pair $(i,j) \in \Omega_{\text{tol}}$, we compute the distances $s_1^{i,j}, s_2^{i,j}$ between the closest surface points on each mesh and apply a soft constraint with threshold $tol$. The loss is formulated as:

$$\mathcal{L}_{\text{pen}} = \gamma_{\text{pen}} \frac{1}{|\Omega_{\text{tol}}|} \sum_{(i,j) \in \Omega_{\text{tol}}} \left( \left| s_1^{i,j} - s_2^{i,j} \right| + \max\left(0, \, tol - \left| s_1^{i,j} - s_2^{i,j} \right| \right) \right), \tag{S10}$$

where $\gamma_{\text{pen}}$ is a high impact factor to strictly enforce separation and eliminate implausible interpenetration.

In addition, we introduce a visibility-aware keypoint loss that adaptively downweights occluded joints. Given instance segmentation masks, each projected joint $j$ is assigned:

$$w_j = \begin{cases} 1, & \text{if joint } j \text{ is visible,} \\ \alpha_{\text{occ}}, & \text{if joint } j \text{ is occluded } (\alpha_{\text{occ}} = 0.1). \end{cases}$$

Let $\{\mathbf{u}_j\}$ and $\{\hat{\mathbf{u}}_j\}$ be the ground-truth and estimated 2D joint positions. We define

$$\mathcal{L}_{\text{std}} = \frac{1}{N} \sum_{j=1}^{N} \|\mathbf{u}_j - \hat{\mathbf{u}}_j\|^2, \qquad \mathcal{L}_{\text{vis}} = \frac{1}{N} \sum_{j=1}^{N} w_j \|\mathbf{u}_j - \hat{\mathbf{u}}_j\|^2.$$

Here, $N$ denotes the total number of 2D keypoints used in the reprojection loss. We then combine them as below.

$$\mathcal{L}_{\mathrm{kp}} = \gamma_{\mathrm{std}}\,\mathcal{L}_{\mathrm{std}} \;+\; \gamma_{\mathrm{vis}}\,\mathcal{L}_{\mathrm{vis}}$$

Although conceptually similar to the visibility loss used in mesh reconstruction, where misclassified silhouette pixels are penalized, this formulation operates on joint reprojection errors rather than mask-pixel discrepancies, yielding improved robustness to occlusion in keypoint alignment.

RoBUDDI estimates each person's 3D pose in multi-person scenario, relative to the camera, providing all necessary geometric information for canonicalization. This effectively sets the camera's extrinsic rotation to $I$ and translation to $\vec{0}$.

The original BUDDI optimization takes approximately 60s per image and peaks at 12.48GB of GPU memory on NVIDIA RTX A6000 GPU (batch size=1). Adding our interpenetration and visibility penalties increases runtime to 77s and peak memory to 15.16GB. All experiments use an interpenetration threshold $tol = 0.02$, a penetration loss weight $\gamma_{\mathrm{pen}} = 15$, an occlusion weight $\alpha_{\mathrm{occ}} = 0.1$, and visibility-aware keypoints and keypoint loss blend factors $\gamma_{\mathrm{std}} = \gamma_{\mathrm{vis}} = 0.5$.

## A.2 CANONICAL PERSPECTIVE-TO-ORTHOGRAPHIC VIEW TRANSFORM (PERS2ORTHO)

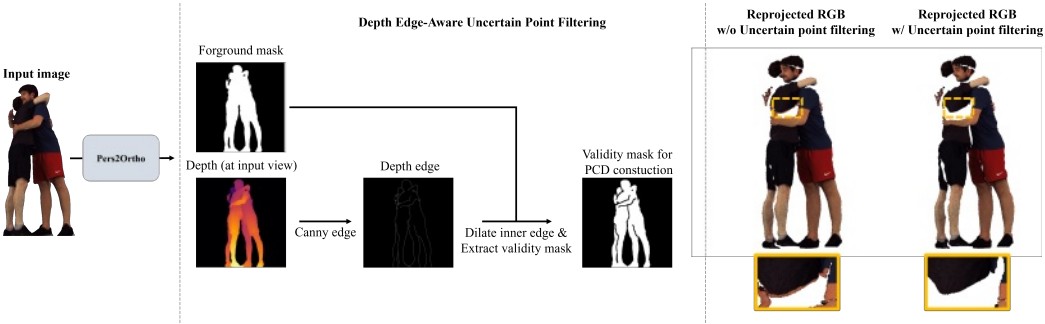

Figure S7: Depth edge-aware filtering removes uncertain boundary regions to improve orthographic projection stability.

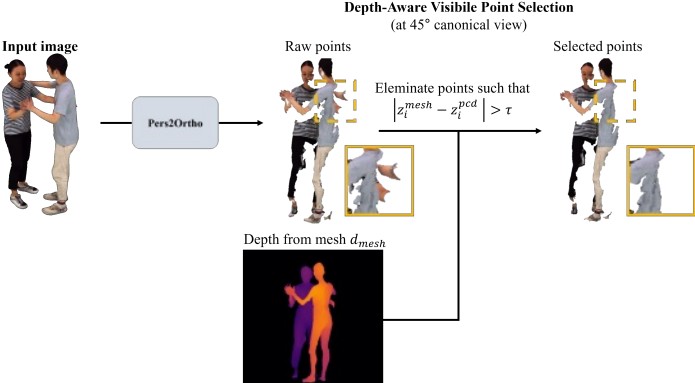

Figure S8: Depth-aware filtering selects front-visible points for clean orthographic projections.

In addition to the primary transformation pipeline described in the main paper, we detail two depth-aware filtering strategies designed to suppress projection artifacts during the conversion from perspective to orthographic views.

**Depth Edge-Aware Uncertain Point Filtering.** As shown in Fig. S7, depth discontinuities often lead to jagged contours and ghosting artifacts near object boundaries. To address this, we detect depth edges using Canny edge detection applied to the rendered depth map. To reduce spurious detections near image borders, we erode the foreground mask before edge extraction. The resulting edge map is then dilated to encompass uncertain boundary regions. A refined validity mask is constructed by

excluding these edge-dilated areas from the projection domain, ensuring that only stable, interior pixels are used in the orthographic projection.

**Depth-Aware Visibile Point Selection.** As represented in Fig. S8, to preserve geometric consistency during the projection of partial point clouds (PCD) into orthographic views, we filter for front-visible points using the rendered depth from the mesh as a geometric prior. This strategy eliminates occluded or background points, retaining only those lying in front of the mesh surface and visible from the target camera view.

Specifically, we project 3D world-space points onto the image plane and sample the mesh depth at corresponding pixel locations. A point is retained if: (1) its projected 2D coordinate lies within image bounds, (2) the absolute depth difference between the point and the mesh is below a threshold $\tau$ (set to $\tau = 0.02$), and (3) the sampled mesh depth is positive.

Formally, let $p_i \in \mathbb{R}^3$ be a 3D point from the PCD, and $z_i^{\text{pcd}}$ and $z_i^{\text{mesh}}$ denote the depths from the point cloud and mesh at the projected location, respectively. The point is retained if:

$$|z_i^{\text{mesh}} - z_i^{\text{pcd}}| < \tau, \quad \text{and} \quad z_i^{\text{mesh}} > 0 \tag{S11}$$

This depth-aware visibility filtering yields cleaner foreground silhouettes and reduces projection noise by removing points that are geometrically inconsistent or lie behind the mesh surface.

We optimize the mesh for partial 3D reconstruction over 200 iterations with a learning rate of 0.02. The Pers2Ortho module, including the reprojection step, takes 16.20 seconds and consumes 14.4GB of VRAM on an NVIDIA A100. We apply a dilation operation with a kernel size of 5 for depth edges.

### A.3    HUMAN GROUP-INSTANCE MULTI-VIEW DIFFUSION (HUG-MVD)

#### A.3.1    TRAINING DATASETS

We leverage one multi-human dataset (Yin et al., 2023) and two single-human datasets (Ho et al., 2023; Yu et al., 2021) to supervise our model with diverse human poses, interactions, and appearances.

**Hi4D** (Yin et al., 2023) is a novel dataset targeting close-range, prolonged human-human interactions with physical contact. Capturing and disentangling such interactions is particularly challenging due to severe occlusions and topological ambiguities. To address this, Hi4D employs individually fitted neural implicit avatars and an alternating optimization scheme that jointly refines surface and pose during close contact. This enables automatic segmentation of fused 4D scans into individual humans. The dataset comprises 100 sequences across 20 subject pairs, totaling over 11K textured 4D scans, all annotated with accurate 2D/3D contact labels and registered SMPL-X models. For our experiments, we extract 1,272 scenes by sampling contact frames with a stride of 16.

**CustomHumans** (Ho et al., 2023) contains high-quality static scans of 80 individuals, captured using a multi-view photogrammetry system with 53 RGB (12MP) and 53 IR (4MP) cameras. Each subject performs a set of predefined motions, including T-pose, hand gestures, and squats, in 10-second sequences at 30 FPS. From each sequence, 4–5 high-fidelity frames are selected, yielding over 600 3D scans. Each sample includes a 40K-face mesh, a 4K texture map, and accurately registered SMPL-X parameters, with a wide range of garment styles (120 total).

**THuman2.0** (Yu et al., 2021) offers 500 high-resolution human scans captured using a dense DSLR rig. Each sample consists of a detailed 3D mesh paired with a high-quality texture map, covering a wide range of body shapes and clothing types. This dataset serves as a clean source of diverse clothed human geometry.

#### A.3.2    RENDERING PROCEDURE

For each 3D human scene, we render 16 views in total, comprising 8 orthographic and 8 perspective images. Each view includes RGB images, normal maps, depth maps, and segmentation masks, rendered from both SMPL-X meshes and original scanned meshes. All views share the same azimuth angles to allow consistent comparison across projection types. Orthographic views are rendered with slight variation in elevation (randomly sampled in the range $[-10°, +10°]$) to improve robustness

against vertical pose and camera variations. Perspective views, on the other hand, utilize elevation values randomly sampled from a broader range of $[-20°, +45°]$ and distances between 2.0 and 6.0 units to enhance variability and generalization.

Camera extrinsic parameters (rotation $\mathbf{R}$ and translation $\mathbf{T}$) are generated using a virtual camera located at the specified distance and orientation, always looking at the origin. These parameters are derived via the `look_at_view_transform` function in PyTorch3D (Ravi et al., 2020).

Prior to rendering, all meshes are normalized to a canonical space. This is done by computing the bounding box of the vertex positions, determining the center and maximal axis length, and scaling the mesh such that it fits within a unit cube. For datasets like CustomHumans (Ho et al., 2023) and THuman2.0 (Yu et al., 2021), we additionally introduce random jittering to the mesh center (on the $X$ and $Y$ axes) to prevent overfitting and encourage generalization. The normalized vertex positions $\mathbf{v}_{\text{norm}}$ are computed by:

$$\mathbf{v}_{\text{norm}} = \frac{\mathbf{v} - \mathbf{c}}{s/2}$$

where $\mathbf{c}$ is the computed center of the bounding box and $s$ is the padded bounding box size.

For orthographic rendering, we fix the camera distance at 3.0 units and maintain consistent scale across all views. For perspective rendering, focal lengths are automatically determined based on the normalized mesh size and camera distance, with additional jitter applied up to 20% to simulate realistic monocular variation.

Rendering is performed using the PyTorch3D `MeshRenderer`, configured with either orthographic or perspective camera models. RGB images are rendered using a Phong shading model with ambient or point lighting depending on the dataset. Depth maps are extracted from the rasterizer's $z$-buffer. Normal maps are generated by interpolating face vertex normals in view space. Instance segmentation masks are computed per-pixel using face-to-instance ID mappings. When contact information is available, contact masks are also rendered for the case of multi-human dataset by identifying faces associated with contact regions and projecting them to image space. Small holes in the resulting binary masks are filled using post-processing.

All outputs are rendered at a base resolution of $768 \times 768$ pixels. For training, we randomly select a reference view and sample six additional views at fixed relative azimuth angles of $\{0°, 45°, 90°, 180°, 270°, 315°\}$, resulting in a 6-view training input for each instance.

### A.3.3 MASKING STRATEGY FOR OCCLUSION SIMULATION

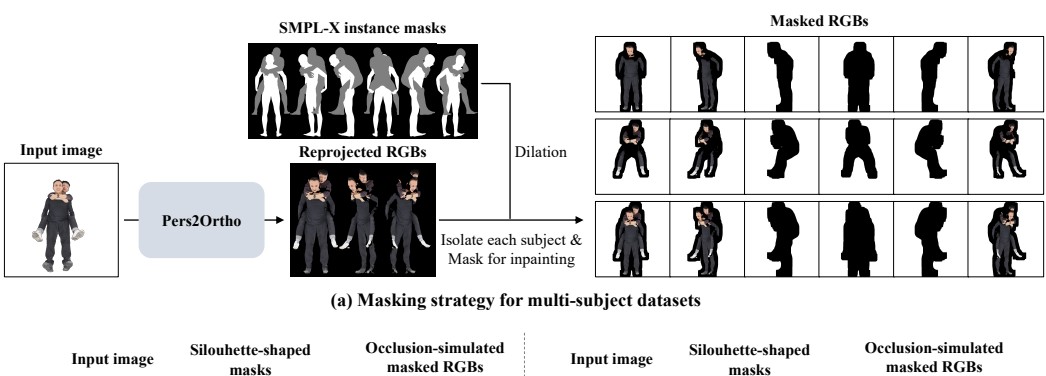

(a) Masking strategy for multi-subject datasets

(b) Masking strategy for single-subject datasets

Figure S9: Illustration of masking strategy for occlusion simulation. (a) For multi-subject datasets, SMPL-X instance masks are used to isolate each subject and specify regions to inpaint. (b) For single-subject datasets, we simulate oclusion through sillouhette-shaped masks.

To simulate realistic visibility and occlusion in multi-human 3D scenes, we construct masked images $x_{\text{mask}}^{(i)}$ from reprojected point clouds $x_{\text{pcd}}^{(i)}$ and visibility masks $M_{\text{vis}}$, which indicate regions of missing observation in each canonical view $\mathcal{C}_i$.

To define the canonical visibility views, we use the reprojected RGB images captured at $\{0°, 45°, 315°\}$. For all other viewpoints, only the SMPL-X instance masks are used for occlusion simulation without relying on PCD.

During training, we generate masked RGB images $\{x_{\text{mask}}^{(i)}\}_{i=0}^5$ to guide inpainting or occlusion-aware reconstruction networks: background regions are labeled as 1, while occluded or missing areas are labeled as 0 and the visible region with the pixel values. These masks provide supervision for learning to reconstruct or inpaint plausible content in the occluded regions.

In multi-subject datasets such as Hi4D (Yin et al., 2023), where mutual occlusion naturally occurs, we use SMPL-X instance masks to isolate each subject. These masks are further dilated to account for peripheral structures such as garments and hair, ensuring that edge regions around the human silhouette are adequately covered for inpainting tasks. We apply a dilation operation with a kernel size of 61.

In single-subject datasets such as CustomHumans (Ho et al., 2023) and THuman2.0 (Yu et al., 2021), we simulate occlusion by randomly selecting one of two masking strategies with equal probability (0.5): (1) silhouette-shaped masks that resemble human figures, randomly scaled and positioned near the image center to mimic the presence of an occluder, or (2) random hole-based masks, such as freeform or template-driven occlusions, which introduce unstructured masking artifacts. This augmentation scheme enables the model to generalize to a wide range of occlusion scenarios, even in the absence of multiple real subjects.

### A.3.4 TRAINING PROCEDURE

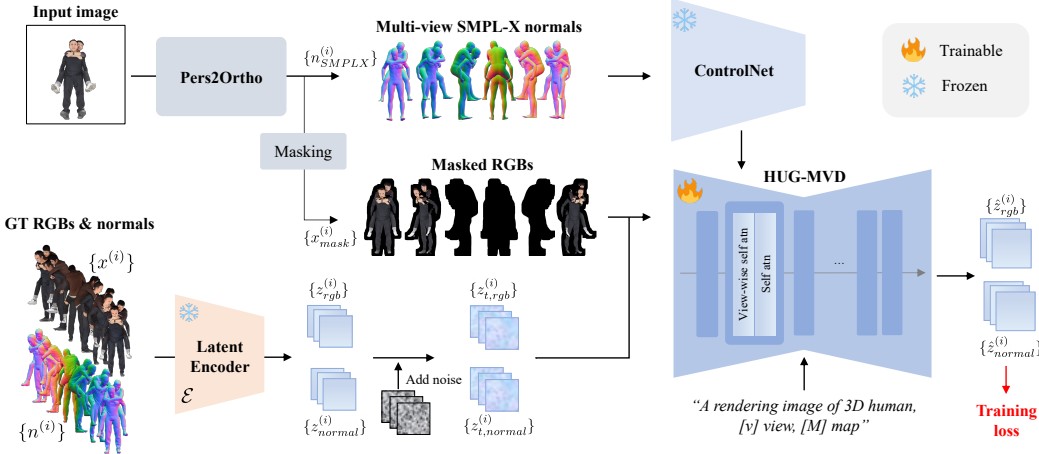

Figure S10: Illustration of the training procedure of HUG-MVD. The model takes occluded RGB images and SMPL-X normal maps from six views and learns to reconstruct complete RGB and normal views via a multi-view diffusion model. SMPL-X guidance is injected via ControlNet, with attention applied across views and modalities for coherent multi-human reconstruction.

The objective of our training procedure is to reconstruct complete RGB and normal images from partially visible, occluded inputs by leveraging a multi-view diffusion model. An overview of the training process is illustrated in Fig. S10. Each training sample consists of six canonical views per scene. The model receives the following inputs: (1) Occluded RGB images reprojected from point clouds using the masking strategies described earlier: $\{x_{\text{mask}}^{(i)}\}_{i=0}^5$, and (2) Corresponding SMPL-X normal maps providing geometric structure: $\{n_{\text{SMPLX}}^{(i)}\}_{i=0}^5$. The supervision targets are: (1) Ground-truth RGB images: $\{x^{(i)}\}_{i=0}^5$, and (2) Ground-truth normal maps: $\{n^{(i)}\}_{i=0}^5$.

We initialize the model from PSHuman (Li et al., 2024b), a pre-trained diffusion model designed to synthesize six RGB and normal views from a single RGB image. Notably, PSHuman is trained exclusively on single-human datasets, including THuman2.0 (Yu et al., 2021) and CustomHumans (Ho et al., 2023), and the open-source version of PSHuman does not support SMPL-X conditioning. To overcome this limitation and extend the model to multi-human scenes with explicit geometry input, we integrate ControlNet (Zhang et al., 2023) into the architecture. This enables the model to utilize SMPL-X normal maps as structural guidance during training.

Our model operates in the latent space defined by the variational autoencoder (VAE) from Stable Diffusion 2.1 (Rombach et al., 2022). Each ground-truth RGB image $x^{(i)}$ and normal map $n^{(i)}$ is encoded into latent variables $z_{\text{rgb}}^{(i)}$ and $z_{\text{normal}}^{(i)}$ using the VAE encoder. The model then predicts the residual noise added to these latents during the forward diffusion process. The training objective is defined as:

$$\mathcal{L}_{\text{diff}} = \sum_{i=0}^{5} \left( \mathbb{E}_{t,\epsilon} \left[ \left\| \epsilon - \epsilon_\theta(z_{t,\text{rgb}}^{(i)}, t, x_{\text{mask}}^{(i)}, n_{\text{SMPLX}}^{(i)}) \right\|_2^2 \right] + \mathbb{E}_{t,\epsilon} \left[ \left\| \epsilon - \epsilon_\theta(z_{t,\text{normal}}^{(i)}, t, x_{\text{mask}}^{(i)}, n_{\text{SMPLX}}^{(i)}) \right\|_2^2 \right] \right) \tag{S12}$$

Here, $z_{t,\text{rgb}}^{(i)}$ and $z_{t,\text{normal}}^{(i)}$ denote the noisy latent variables at timestep $t$, and $\epsilon$ is the Gaussian noise sampled during training. The denoising model $\epsilon_\theta$ learns to recover the clean signal conditioned on the masked RGB inputs and the SMPL-X geometry.

Two key attention modules are employed to support effective multi-view generation. First, following prior work (Li et al., 2024a;b), we apply row-wise multi-view self-attention independently within each modality (RGB or normal), allowing the model to correlate across the six canonical views. Second, to enable information exchange between RGB and normal modalities, we apply self-attention across all latent tokens, allowing cross-modality attention between RGB and normal features at each view. This joint RGB-normal attention mechanism is also inspired by (Li et al., 2024a;b).

During training, only the U-Net parameters are optimized, while the remaining components such as the CLIP image encoder and the VAE are kept frozen. Text prompt embeddings are injected via cross-attention using the template: "`a rendering image of 3D human, [V] view, [M] map`", where [V] indicates the view direction (e.g., `front`, `left`, `face`) and [M] specifies the modality (`color` or `normal`).

We also experimented with classifier-free guidance (CFG) by applying conditioning dropout, following prior work (Ho & Salimans, 2021). However, since CFG yielded only marginal performance improvements at test time, it was not used during inference.

To address occlusion and mesh collision issues in multi-human scenes, we optionally apply contact-aware binary masks on Hi4D (Yin et al., 2023) samples. These masks suppress gradient updates in regions of body-to-body contact where the mesh supervision is less reliable due to penetration artifacts or annotation noise.

The model is trained on a single NVIDIA A100 GPU (80GB) with a batch size of 16 and gradient accumulation steps of 8. We use the Adam (Kingma & Ba, 2014) optimizer with a learning rate of $5 \times 10^{-6}$, $\beta_1 = 0.9$, and $\beta_2 = 0.999$. Training follows a two-stage curriculum: (1) pre-training without inpainting masks for 1,000 steps, followed by (2) fine-tuning with inpainting masks for another 1,000 steps to simulate occlusion. The entire training takes approximately two days, with peak GPU memory usage of around 62GB. A DDPM scheduler with 1,000 diffusion steps is used throughout training.

### A.3.5 INFERENCE PROCEDURE

At inference time, the model predicts complete normal maps $\{\hat{n}^{(i)}\}$ and synthesizes complete RGB images $\{\hat{x}^{(i)}\}$ across all views from partially visible RGB inputs and canonical-view SMPL-X normal maps as illustrated in Fig. S11.

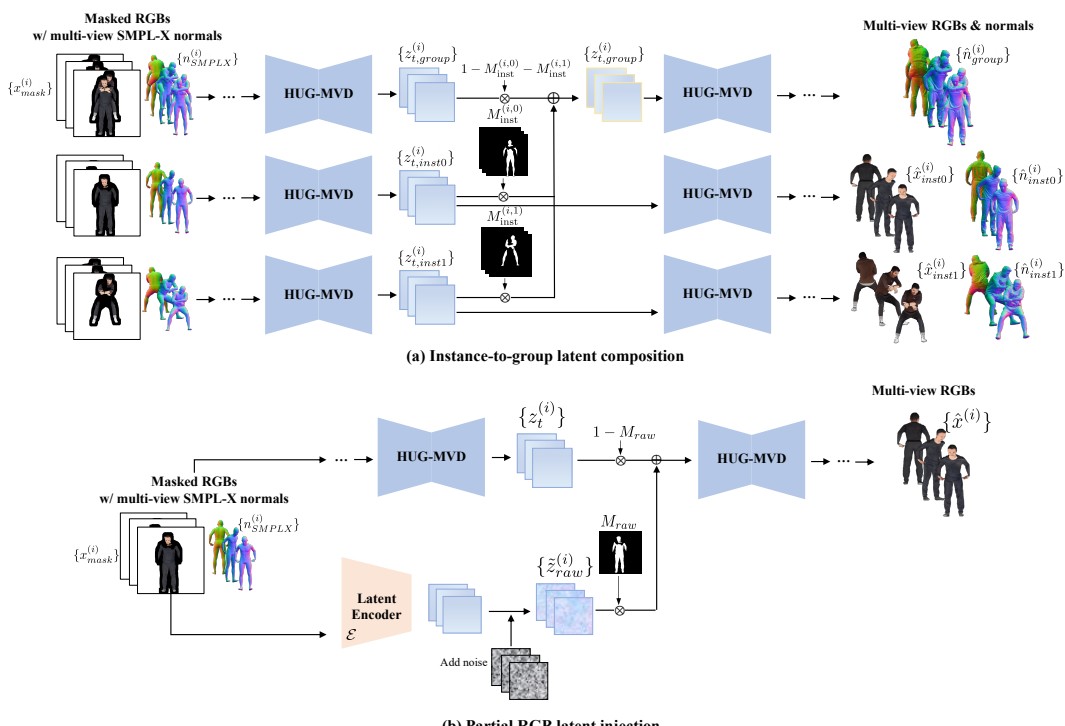

Figure S11: Illustration of the inference procedure of HUG-MVD. (a) Instance-to-group latent composition and (b) partial RGB latent injection for RGB synthesis.

To maintain consistency between group-level and instance-level reconstructions, we perform latent-space composition. At each diffusion timestep $t$, instance-specific latents $z_{t,\text{inst}(k)}^{(i)}$ are softly injected into the group-level latent $z_{t,\text{group}}^{(i)}$ using binary masks $\mathbb{1}_{\text{inst}}^{(i,k)}$ and a blending ratio $\alpha_{\text{gi}} = 0.8$:

$$z_{t,\text{group}}^{(i)} \leftarrow \sum_{k=1}^{K} \left[ \alpha_{\text{gi}} \cdot \mathbb{1}_{\text{inst}}^{(i,k)} \cdot z_{t,\text{inst}(k)}^{(i)} + (1 - \alpha_{\text{gi}}) \cdot \mathbb{1}_{\text{inst}}^{(i,k)} \cdot z_{t,\text{group}}^{(i)} \right] + \left( 1 - \sum_{k=1}^{K} \mathbb{1}_{\text{inst}}^{(i,k)} \right) \cdot z_{t,\text{group}}^{(i)}, \quad \text{(S13)}$$

This mechanism allows high-frequency details from instance-specific predictions to be integrated into the global group representation, improving surface continuity in multi-human scenes.

Also, to further enhance RGB quality, we inject latent signals from partially visible RGB inputs. At each diffusion timestep $t$, we generate a noisy version $\tilde{z}_{\text{raw}}$ of the raw RGB latent (restricted to visible regions) and blend it into the current latent $z_t$ using a binary mask $\mathbf{m}_{\text{raw}}$ and mixing ratio $\alpha_{\text{pcd}} = 0.8$:

$$z_t \leftarrow \mathbf{m}_{\text{raw}} \cdot \left[ \alpha_{\text{pcd}} \cdot \tilde{z}_{\text{raw}} + (1 - \alpha_{\text{pcd}}) \cdot z_t \right] + (1 - \mathbf{m}_{\text{raw}}) \cdot z_t, \quad \text{(S14)}$$

This operation is applied exclusively to the RGB branch and aims to reinforce reliable visual priors in visible areas, improving fidelity in occluded or ambiguous regions. We apply this injection selectively to low-confidence views (e.g., non-source views) to avoid overwriting already plausible outputs.

We use a DDIM scheduler with $\eta = 1.0$ and perform 40 denoising steps per sample. We also used $\alpha_{\text{gi}} = \alpha_{\text{pcd}} = 0.8$. Inference for all group-level and instance-level multi-view RGB and normal maps takes 60.16 seconds, using 34.76GB of VRAM on an NVIDIA A100.

## A.4 HUMAN GROUP-INSTANCE GEOMETRY RECONSTRUCTION (HUG-GR)

Here, we provide a detailed explanation of the two geometry-level supervision terms—*interpenetration loss* and *visibility loss*. As illustrated in Fig. S12, these losses play

complementary roles in enhancing geometric plausibility and part-level visibility consistency during group-instance reconstruction.

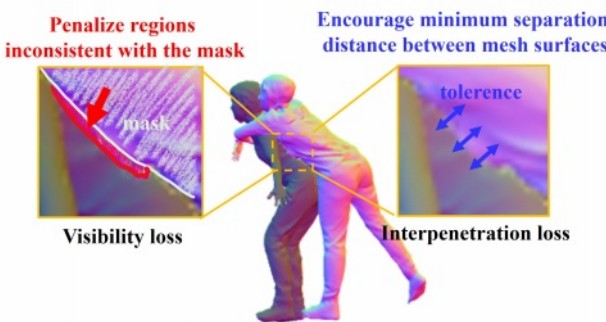

Figure S12: Illustration of the interpenetration loss and visibility loss. The interpenetration loss penalizes body-part collisions (blue arrows), while the visibility loss enforces alignment between rendered masks and ground-truth visibility (red region).

**Interpenetration Loss.** To prevent anatomically implausible overlaps between articulated body parts, we define an interpenetration loss that penalizes violations among predefined part pairs $(i, j) \in \Omega_{\text{tol}}$, where $\Omega_{\text{tol}}$ is the tolerance set encoding pairs subject to collision constraints inspired by (Huang et al., 2024). To determine the tolerance set, $\Omega_{tol}$, we first compute a contact map based on vertex-to-vertex distances on the initial SMPL-X meshes, which identifies all potential contact regions. The resulting contact regions, constitutes our tolerance set. For each pair, we compute the shortest distances $s_1^{i,j}$ and $s_2^{i,j}$ between the nearest surface points of parts $i$ and $j$. These distances are then softly constrained using a tolerance threshold $tol$. Our formulation assigns a fixed penalty value of tol for any distance within this threshold. This acts as a step-like penalty that strongly discourages any violation of the minimum distance, rather than penalizing the magnitude of the penetration. While this approach does not provide a gradient within the penetrated region, it can enhance training stability by bounding the penalty for any single pair.

$$\mathcal{L}_{\text{pen}} = \frac{1}{|\Omega_{\text{tol}}|} \sum_{(i,j) \in \Omega_{\text{tol}}} \left( |s_1^{i,j} - s_2^{i,j}| + \max\left(0, \ tol - |s_1^{i,j} - s_2^{i,j}|\right) \right) \tag{S15}$$

This term encourages a minimum surface separation between adjacent parts (e.g., thighs vs. calves), helping to reduce self-penetration artifacts while preserving flexibility for naturally close configurations such as seated or folded poses.

**Visibility Loss.** To improve spatial alignment in crowded scenes, we supervise visibility using rendered segmentation masks. For each body part $b$ in instance $k$, we penalize visibility mismatches using:

$$\mathcal{L}_{\text{vis}} = \frac{1}{2B} \sum_{k=1}^{K} \sum_{b=1}^{B} \frac{E_b^k}{M_b^k + \epsilon}, \tag{S16}$$

where $E_b^k$ is the number of incorrectly occluded pixels and $M_b^k$ the total visible pixels in the ground truth. This encourages accurate silhouette and occlusion boundaries, particularly in group interactions.

**Adaptive Region-Specific Optimization.** To balance global stability and preservation of local details, we apply region-specific optimization strategies. In particular, lower learning rates are used for vertices located in semantically and geometrically complex regions such as the hands and face. This allows the model to better preserve high-frequency features provided in the initial SMPL-X mesh in these areas while still allowing flexible carving of geometric features, such as clothing, in other regions. We determine how close a vertice is to a complex region using the optimized SMPL-X joint positions of the hands and face. And use sigmoid blending to derive the actual learning rate. Formally, the vertice-wise adaptive learning rate $\alpha_{\text{v}}$ for vertex $v$ is:

$$\alpha_{\text{v}} = \alpha_{\text{base}} \cdot \frac{1}{e^{-(200d_v + 10)} + 1}, \tag{S17}$$

Where $\alpha_{\text{base}}$ is the base learning rate and $d_v$ is the minimum distance between $v$ and the set of all SMPL-X joint vertices in consideration denoted as $J_{\text{SMPL-X}}$. Then, $d_v$ is:

$$d_v = \min_{j \in J_{\text{SMPL-X}}} \|v - j\| \tag{S18}$$

Thus, we assign lower learning rates to vertices with smaller $d_v$ (i.e. vertices closer to hands or the face). As illustrated in Fig. 6(b), this adaptive strategy results in sharper reconstructions of fine regions (e.g., fingers, facial contours) while maintaining coherence in broader anatomical parts like the torso or limbs.

We optimize the mesh over 200 iterations with a learning rate of 0.01, $\lambda_{\text{group}} = 1.0$, $\lambda_{\text{inst}} = 0.2$, $\lambda_{\text{pen}} = 2.0$ and $\lambda_{\text{vis}} = 1.0$. HUG-GR takes 125.47 seconds and consumes 7.58GB of VRAM on an A100 GPU.

## A.5 Occlusion- and View-Aware Texture Fusion

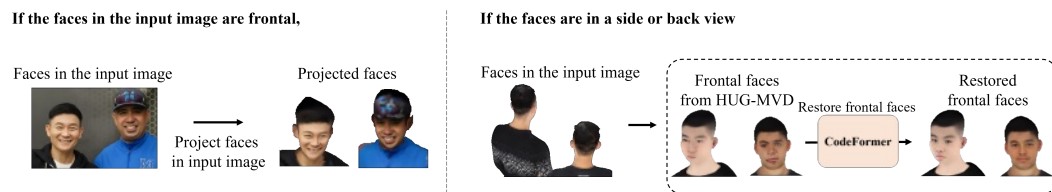

Figure S13: View-aware face restoration enhances frontal views using landmark-guided inpainting.

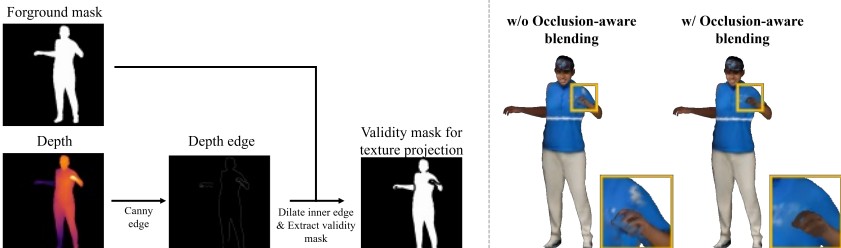

Figure S14: Illustration of the occlusion-aware blending strategy. Edge-aware confidence masks, computed from depth discontinuities, suppress artifacts near occlusion boundaries, resulting in cleaner silhouettes and improved cross-view consistency.

To construct coherent and high-quality full-body textures, we fuse multi-view RGB predictions into a unified texture. To improve texture fidelity and suppress artifacts, we introduce two important enhancements: *view-aware face restoration* and *occlusion-aware blending*.

**View-Aware Face Restoration.** Faces captured from extreme angles or under occlusion often exhibit degraded appearance. To address this, as shown in Fig. S13, we first analyze each view using facial landmarks and SMPL-X head orientation to estimate the relative frontalness of the face. Among the six RGB predictions per instance, we select the two most frontal views. If the source view is used for the face, we directly use its content. If not, we perform face inpainting on the most frontal view using CodeFormer (Zhou et al., 2022), where a soft circular mask is generated using warped 5-point facial landmarks. In both cases, the enhanced face region is warped back and blended into the original view using inverse affine transformation. This step improves the final texture synthesis especially in the face region.

**Occlusion-Aware Blending.** As illustrated in Fig. S14, to prevent ghosting and bleeding artifacts near occlusion boundaries, we employ edge-aware confidence masking guided by view-dependent depth maps. Depth edges are first extracted using the Canny filter, and the resulting edge map is dilated with a fixed kernel to define an exclusion zone. We retain only the pixels that lie within foreground regions and are sufficiently distant from detected depth discontinuities. These reliable pixels are used to generate a binary confidence mask $C_i$ for each view. The final contribution of a view's texture projection $T_i$ is modulated by this mask as $T_i' = C_i \cdot T_i$. This occlusion-aware blending

strategy effectively suppresses unstable regions near self-occlusion edges, resulting in cleaner object silhouettes and improved consistency across views. We apply a dilation operation with a kernel size of 21.

Our texture fusion takes 14.49 seconds and consumes 4.95GB of VRAM on an A100 GPU.

## B  EVALUATION DETAILS

### B.1  EVALUATION SETTINGS

We evaluate our method in comparison to prior works across three categories: methods of single human reconstruction from single image, methods of multi-human reconstruction from multi-view images, and methods of multi-human reconstruction from videos. Since there is no publicly available baseline implementation that directly performs multi-human reconstruction from a single image as represented in Tab. S5, To ensure a fair comparison, we follow the evaluation protocol of (Cha et al., 2024) by adapting related methods in each category under consistent settings. To isolate the effect of SMPL-X prediction from the reconstruction process, all main comparisons are conducted using ground-truth SMPL-X. Results using predicted SMPL-X are included in Sec. C.1 of the supplementary.

Table S5: Comparison of recent 3D human reconstruction methods. HUG3D supports multi-human reconstruction from a single image with both geometry and texture.

| Method | Multi- or Single Human | Input Type | Geometry | Texture | Publicly Available |
|---|---|---|---|---|---|
| ECON (Xiu et al., 2023) | Single human | Single image | ✓ | ✗ | ✓ |
| SiTH (Ho et al., 2024) | Single human | Single image | ✓ | ✓ | ✓ |
| SIFU (Zhang et al., 2024) | Single human | Single image | ✓ | ✓ | ✓ |
| PSHuman (Li et al., 2024b) | Single human | Single image | ✓ | ✓ | ✓ |
| DeepMultiCAP (Zheng et al., 2021) | Multi-human | Multi-view images | ✓ | ✗ | ✓ |
| Multiply (Jiang et al., 2024) | Multi-human | Video | ✓ | ✓ | ✓ |
| Cha et al. (Cha et al., 2024) | Multi-human | Single image | ✓ | ✗ | ✗ |
| **HUG3D (Ours)** | Multi-human | Single image | ✓ | ✓ | ✓ (upon acceptance) |

**Single human reconstruction from single image.** For methods of originally designed for single human reconstruction (Zhang et al., 2024; Ho et al., 2024; Li et al., 2024b; Xiu et al., 2023), we adapted them to the multi-human setting as follows. Ground-truth instance segmentation masks were used to isolate each person in the input image. Each individual was then reconstructed independently using the corresponding method. Since the outputs lie in different coordinate frames, we performed a canonicalization procedure to align all reconstructions into a shared space. Specifically, for each instance, we first predicted the SMPL-X mesh using the method's native estimator. We then computed a similarity transformation—comprising scale, rotation, and translation—that aligns the ground-truth SMPL-X mesh to the predicted one. The ground-truth SMPL-X mesh was transformed into the predicted space before reconstruction, and the reconstruction output was transformed back to the ground-truth space via the inverse transformation, allowing the reconstructed scene to be composited consistently. We also evaluate PSHuman-multi, which applies the single-person reconstruction pipeline PSHuman (Li et al., 2024b) directly to uncropped multi-person images. Since we use ground-truth SMPL-X for all evaluations, we omit the SMPL-X optimization process for baselines that originally involve it.

**Multi-human reconstruction from multi-view image.** For multi-view baselines (Zheng et al., 2021), we provided only a single view as input for inference, to ensure comparability with our single-image reconstruction setting.

**Multi-human reconstruction from videos.** Similarly, for video-based baselines (Jiang et al., 2024), we provided only a single image as the first frame for inference.

### B.2  EVALUATION DATASET

**MultiHuman (Jiang et al., 2024).** To facilitate both quantitative and qualitative evaluation of reconstructed meshes, we rendered perspective-view images from the MultiHuman dataset using a multi-view setup. Our evaluation covers a total of 20 two-person scenes, including 6 closely interactive cases (sequences 8, 23, 24, 250, 252, 253) and 14 naturally interactive scenes (sequences 12, 16, 17, 18, 19, 20, 22, 30, 226, 244, 249, 251, 255, 256). For ablation studies, we focus on the closely interactive cases.

For each scene, we rendered the meshes from 4 distinct camera viewpoints, generated by sampling a random azimuth and adding fixed offsets of $\{0°, 90°, 180°, 270°\}$, resulting in views uniformly distributed around the subject. The elevation angles were randomly sampled in the range $[-20°, 45°]$, and the camera-to-subject distances were sampled uniformly from $[2.0, 6.0]$, simulating varying levels of zoom and perspective distortion.

To ensure scale-invariant and consistently framed rendering, each mesh was normalized to fit within a unit cube centered at the origin. This was achieved by computing the mesh's axis-aligned bounding box and uniformly scaling it based on the maximum side length.

**In-the-wild.** For our qualitative evaluation with in-the-wild images, we leveraged OpenAI's Sora service to obtain a diverse set of test images. Sora performs a web-based image search for user-specified concepts, reconstructs novel scenes by referencing those search results, and synthesizes new images that reflect real-world variation. The resulting Sora outputs—whose content is derived from Internet-sourced photos—were then used as our "in-the-wild" evaluation set, ensuring that our method is tested on unconstrained, naturally diverse imagery.

## B.3 EVALUATION METRICS

We employ a comprehensive set of metrics to evaluate both the geometric and texture quality of reconstructed multi-human meshes. These metrics cover surface accuracy, physical realism, and perceptual quality. Here, P and Q are point clouds sampled from the predicted and ground-truth meshes.

**Chamfer Distance (CD).** Chamfer Distance measures the bidirectional discrepancy between the predicted and ground-truth surfaces. We uniformly sample 100,000 points from each mesh surface and compute the average closest-point distance from the predicted points to the ground-truth surface and vice versa. The final CD score is defined as:

$$\text{CD}(P, Q) = \frac{1}{|P|} \sum_{p \in P} \min_{q \in Q} \|p - q\|_2 + \frac{1}{|Q|} \sum_{q \in Q} \min_{p \in P} \|q - p\|_2.$$

A lower CD indicates a more accurate reconstruction that closely matches the geometry of the ground truth in both completeness and precision.

**Point-to-Surface Distance (P2S).** P2S measures the unidirectional accuracy of the predicted surface with respect to the ground-truth shape. Specifically, it measures the average Euclidean distance from each point sampled on the predicted mesh to the closest point on the ground-truth surface:

$$\text{P2S}(P \rightarrow Q) = \frac{1}{|P|} \sum_{p \in P} \min_{q \in Q} \|p - q\|_2^2.$$

P2S emphasizes surface accuracy without penalizing missing parts, and lower values indicate closer alignment to the reference shape.

**Normal Consistency (NC).** NC measures the angular similarity between surface normals on the predicted and ground-truth meshes. For each point, we compare the normal vector at that point with the normal vector at the closest point on the opposite surface. The final score is averaged bidirectionally:

$$\text{NC}(P, Q) = \frac{1}{2|P|} \sum_{p \in P} \left(1 - \langle \mathbf{n}_p, \mathbf{n}_{\text{NN}(p,Q)} \rangle\right) + \frac{1}{2|Q|} \sum_{q \in Q} \left(1 - \langle \mathbf{n}_q, \mathbf{n}_{\text{NN}(q,P)} \rangle\right),$$

where $\langle \cdot, \cdot \rangle$ denotes the dot product between unit normals, and $\text{NN}(\cdot)$ returns the nearest neighbor in the opposite set. A higher NC indicates better preservation of surface orientations and local detail.

**F-score.** F-score evaluates both precision and recall of the predicted surface points with respect to a ground-truth reference under a distance threshold $\tau$. We use $\tau = 1\text{cm}$. Precision measures the percentage of predicted points that lie within $\tau$ of the ground-truth surface, while recall measures the converse. F-score is defined as the harmonic mean of the two:

$$\text{F-score} = \frac{2 \cdot \text{Precision} \cdot \text{Recall}}{\text{Precision} + \text{Recall}}.$$

This metric rewards reconstructions that are both accurate and complete.

**Bounding Box IoU (bbox-IoU).** We compute the 3D Intersection-over-Union (IoU) of axis-aligned bounding boxes of the predicted and ground-truth meshes:

$$\text{IoU}_{\text{bbox}} = \frac{\text{vol}(B_{\text{pred}} \cap B_{\text{gt}})}{\text{vol}(B_{\text{pred}} \cup B_{\text{gt}})},$$

where $B_{\text{pred}}$ and $B_{\text{gt}}$ are the predicted and ground-truth bounding boxes, respectively. This metric evaluates global layout similarity and spatial coverage.

**L2 Normal Error.** We assess surface detail preservation by computing the per-pixel $L_2$ distance between rendered normal maps of the predicted and ground-truth meshes. This is done across four orthographic views at azimuth angles $\{0°, 90°, 180°, 270°\}$:

$$\text{L2-NormErr} = \frac{1}{N} \sum_{i=1}^{N} \|\mathbf{n}_i^{\text{pred}} - \mathbf{n}_i^{\text{gt}}\|_2^2.$$

We also report this error computed within occluded regions, to specifically assess reconstruction quality under visual occlusion.

**Contact Precision (CP).** To evaluate the physical plausibility of multi-human reconstruction, we measure the alignment of predicted inter-body contact regions with the ground truth. This metric quantifies how accurately the predicted contact points reflect the true contact between two human bodies.

Let $\hat{M}_1$ and $\hat{M}_2$ be the predicted meshes, and $M_1$ and $M_2$ the corresponding ground-truth meshes. Denote their vertex sets as $\hat{V}_1$, $\hat{V}_2$, $V_1$, and $V_2$, respectively. A vertex is considered in contact if it lies within a threshold distance $\delta$ from the other mesh.

First, we define the ground-truth contact region $\mathcal{C}_{\text{gt}}$ as:

$$\mathcal{C}_{\text{gt}} = \left\{ v \in V_1 \cup V_2 \,\middle|\, \min_{v' \in V_2 \cup V_1} \|v - v'\|_2 < \delta \right\},$$

and similarly, the predicted contact region $\mathcal{C}_{\text{pred}}$ as:

$$\mathcal{C}_{\text{pred}} = \left\{ \hat{v} \in \hat{V}_1 \cup \hat{V}_2 \,\middle|\, \min_{\hat{v}' \in \hat{V}_2 \cup \hat{V}_1} \|\hat{v} - \hat{v}'\|_2 < \delta \right\}.$$

We then compute precision by counting the proportion of predicted contact points that are close to the ground-truth contact region:

$$\text{CP} = \frac{1}{|\mathcal{C}_{\text{pred}}|} \sum_{\hat{v} \in \mathcal{C}_{\text{pred}}} \mathbf{1}\left[\text{NN}(\hat{v}, V_{\text{gt}}) \in \mathcal{C}_{\text{gt}}\right],$$

where $\text{NN}(\hat{v}, V_{\text{gt}})$ denotes the nearest vertex to $\hat{v}$ among all ground-truth vertices.

We set the contact threshold $\delta = 0.01$ meter. A higher CP indicates better prediction of physically plausible inter-human contacts.

**Texture Fidelity.** To assess the perceptual quality of the reconstructed texture, we evaluate the rendered mesh images against ground-truth renderings using three standard image similarity metrics: PSNR, SSIM, and LPIPS.

Given the predicted image $\hat{I}$ and the ground-truth image $I$ rendered from the same view, we compute:

**Peak Signal-to-Noise Ratio (PSNR)**:

$$\text{PSNR}(I, \hat{I}) = 10 \cdot \log_{10}\left(\frac{(L_{\max})^2}{\text{MSE}(I, \hat{I})}\right),$$

where $L_{\max} = 255$ and MSE denotes the mean squared error between pixel values. A higher PSNR indicates better reconstruction.

**Structural Similarity Index Measure (SSIM).**

$$\text{SSIM}(I, \hat{I}) = \frac{(2\mu_I \mu_{\hat{I}} + c_1)(2\sigma_{I\hat{I}} + c_2)}{(\mu_I^2 + \mu_{\hat{I}}^2 + c_1)(\sigma_I^2 + \sigma_{\hat{I}}^2 + c_2)},$$

where $\mu$, $\sigma^2$, and $\sigma_{I\hat{I}}$ denote means, variances, and covariances of local patches. SSIM captures perceptual similarity in terms of luminance, contrast, and structure.

**Learned Perceptual Image Patch Similarity (LPIPS).** LPIPS compares features from a pretrained deep network (e.g., AlexNet) between $\hat{I}$ and $I$, and correlates better with human judgment of perceptual similarity. Lower LPIPS indicates better quality.

These metrics are computed over four rendered views $\{0°, 90°, 180°, 270°\}$, under orthographic projection. We also report masked versions of these metrics that are evaluated only on occluded foreground regions, allowing more fine-grained assessment under challenging interaction scenarios.

**Occlusion-aware Metrics.** To evaluate reconstruction quality under challenging visibility conditions, we compute occlusion-aware variants of image-based metrics and surface normal metrics by restricting the evaluation to regions occluded by another human instance.

Let $I^{(i)}$ and $\hat{I}^{(i)}$ denote the ground-truth and predicted images of instance $i \in \{0, 1\}$, and let $M^{(j)}$ be the binary mask of the other instance $j \neq i$. A pixel $(x, y)$ is considered occluded in instance $i$ if it belongs to $M^{(j)}$ and the corresponding pixel in $I^{(i)}$ is not background (i.e., not white):

$$\mathcal{O}^{(i)} = \left\{ (x, y) \,\middle|\, M^{(j)}(x, y) = 1 \ \wedge \ I^{(i)}(x, y) \neq background \right\}.$$

We then compute each metric by applying the occlusion mask $\mathcal{O}^{(i)}$ to both predicted and ground-truth images:

$$\text{Occ-PSNR}^{(i)} = \text{PSNR}\left( \hat{I}^{(i)}|_{\mathcal{O}^{(i)}}, I^{(i)}|_{\mathcal{O}^{(i)}} \right),$$

$$\text{Occ-SSIM}^{(i)} = \text{SSIM}\left( \hat{I}^{(i)}|_{\mathcal{O}^{(i)}}, I^{(i)}|_{\mathcal{O}^{(i)}} \right)$$

For surface normal comparisons, let $N^{(i)}$ and $\hat{N}^{(i)}$ be the ground-truth and predicted normal maps of instance $i$. The occlusion-aware $L_2$ Normal Error is defined as:

$$\text{Occ-L2-NormErr}^{(i)} = \frac{1}{|\mathcal{O}^{(i)}|} \sum_{(x,y)\in\mathcal{O}^{(i)}} \left\| \hat{N}^{(i)}(x, y) - N^{(i)}(x, y) \right\|_2^2.$$

All occlusion-aware metrics are averaged over both instances and across the four canonical viewpoints to provide a robust estimate of reconstruction performance in visually occluded regions.

## C  ADDITIONAL RESULTS OF 3D MULTI-HUMAN RECONSTRUCTION

### C.1  QUALITATIVE COMPARISON INCLUDING ADDITIONAL BASELINES

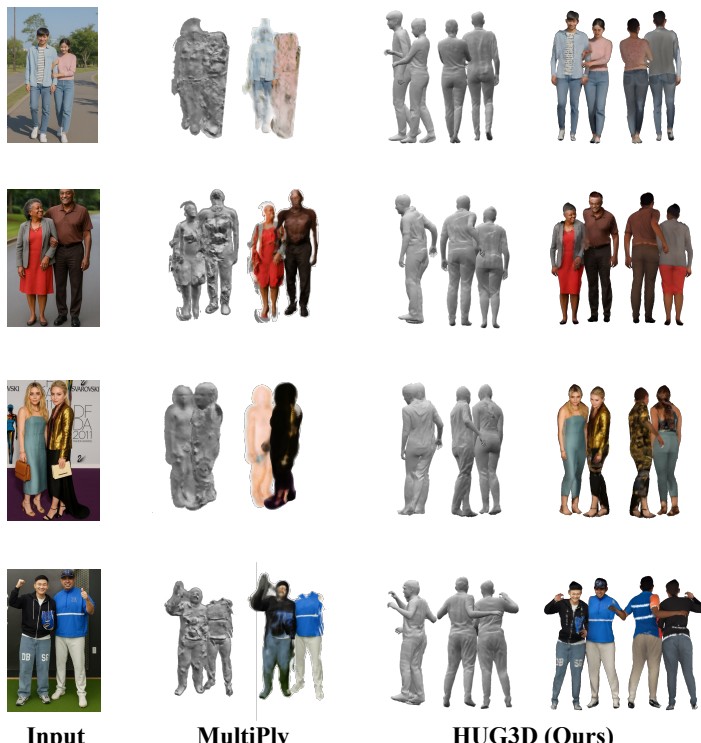

**Input**         **MultiPly**         **HUG3D (Ours)**

Figure S15: Additional qualitative comparison on multi-human 3D reconstruction from a single in-the-wild image. HUG3D outperforms baselines by correcting perspective distortion, preserving inter-human contact, and hallucinating plausible textures under heavy occlusion.

In addition to the baselines presented in the main paper, we include two additional baselines for comparison: DeepMultiCap (Zheng et al., 2021), a method designed for multi-human reconstruction from multi-view images, and Multiply (Jiang et al., 2024), a method for multi-human reconstruction from videos. Fig. S15 presents additional qualitative comparisons on the in-the-wild images, while Fig. S16 shows results on the MultiHuman dataset. In the in-the-wild setting, where SMPL-X predictions are used instead of ground-truth, our method continues to produce high-quality reconstructions, demonstrating robustness to SMPL-X estimation errors. Across all baselines, we observe common failure modes: incomplete geometry and missing textures in occluded regions, severe interpenetration or failure to preserve contact due to the lack of inter-person modeling, and inability to correct perspective distortion in images with complex viewpoints. In contrast, HUG3D consistently delivers robust multi-human reconstructions that preserve contact, correct geometric distortion, and hallucinate plausible textures even under severe occlusion.

### C.2  SEPARATE RESULTS FOR EACH INSTANCE

Table S6: Quantitative comparison of geometry and texture for each instance

| Method | CD ↓ | P2S ↓ | NC ↑ | F-score ↑ | bbox-IoU ↑ | Norm $L2$ ↓ | PSNR ↑ | SSIM ↑ | LPIPS ↓ |
|---|---|---|---|---|---|---|---|---|---|
| SIFU | 6.367 | 2.292 | 0.753 | 30.203 | 0.659 | 0.018 | 17.222 | 0.882 | 0.127 |
| SiTH | 9.642 | 3.166 | 0.712 | 21.740 | 0.541 | 0.024 | 16.090 | 0.881 | 0.143 |
| PSHuman | 16.876 | 6.384 | 0.614 | 9.561 | 0.402 | 0.039 | 13.720 | 0.857 | 0.188 |
| DeepMultiCap | 13.314 | 2.952 | 0.754 | 18.898 | 0.442 | 0.026 | 15.25 | 0.880 | 0.161 |
| **Ours** | **3.531** | **1.719** | **0.816** | **42.946** | **0.801** | **0.012** | **18.659** | **0.894** | **0.102** |

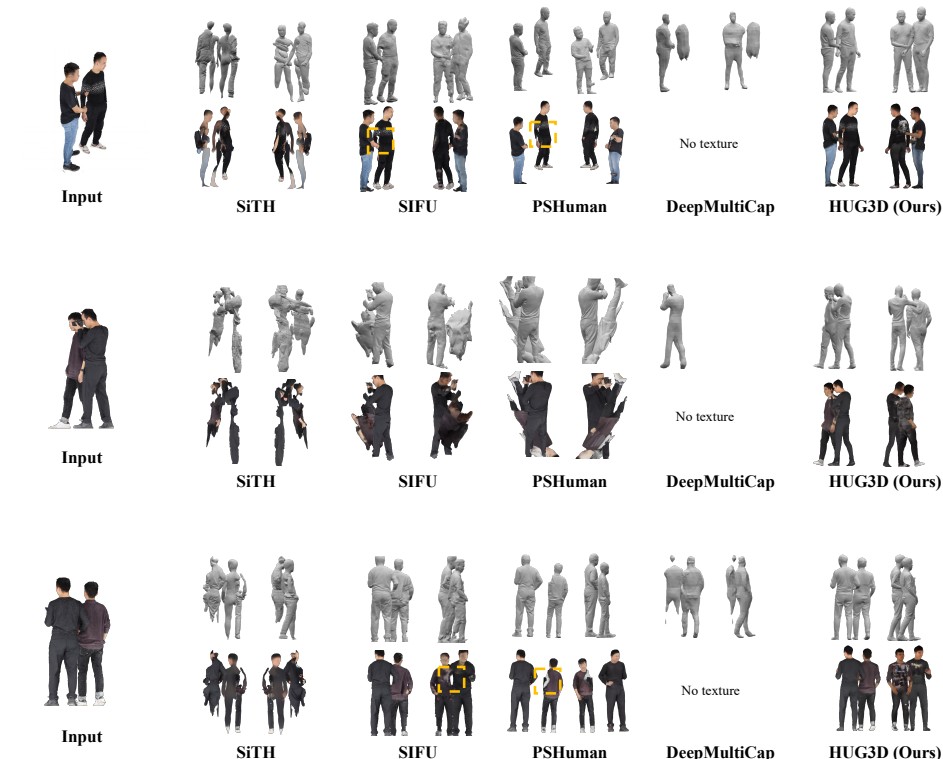

Figure S16: Additional qualitative comparison on multi-human 3D reconstruction from a single image in the MultiHuman dataset. Yellow boxes highlight broken geometry, missing texture, and incorrect inter-human interactions. HUG3D outperforms baselines by correcting perspective distortion, preserving inter-human contact, and hallucinating plausible textures under heavy occlusion.

Table S6 shows per-instance comparisons of geometry and texture metrics. Our method consistently outperforms baselines across all measures, achieving better geometric accuracy (e.g., lowest CD, P2S, and Norm $L2$; highest NC and F-score) and texture quality (highest PSNR/SSIM, lowest LPIPS). This instance-level analysis further highlights the effectiveness of our unified framework in capturing both fine-grained geometry and high-quality appearance.

## C.3 RESULTS DEPENDING ON LEVEL OF INTERACTION

Table S7: Quantitative comparison of geometry depending on level of interaction.

| Interaction | Method | CD ↓ | P2S ↓ | NC ↑ | F-score ↑ | bbox-IoU ↑ | Norm $L2$ ↓ | CP ↑ |
|---|---|---|---|---|---|---|---|---|
| Closely | SIFU | 7.267 | 2.750 | 0.724 | 24.335 | 0.757 | 0.033 | 0.117 |
| | SiTH | 10.908 | 3.491 | 0.697 | 19.216 | 0.694 | 0.044 | 0.281 |
| | PSHuman | 14.920 | 5.518 | 0.616 | 10.572 | 0.631 | 0.065 | 0.049 |
| | DeepMultiCap | 9.6697 | 2.745 | 0.764 | 20.471 | 0.606 | 0.039 | 0.123 |
| | **Ours** | **4.315** | **2.121** | **0.811** | **37.243** | **0.838** | **0.022** | **0.326** |
| Natural | SIFU | 4.895 | 2.069 | 0.768 | 31.510 | 0.788 | 0.026 | 0.076 |
| | SiTH | 8.486 | 3.044 | 0.714 | 21.877 | 0.715 | 0.038 | 0.068 |
| | PSHuman | 15.884 | 6.350 | 0.617 | 9.370 | 0.671 | 0.070 | 0.017 |
| | DeepMultiCap | 17.081 | 2.463 | 0.741 | 17.018 | 0.470 | 0.052 | 0.064 |
| | **Ours** | **3.340** | **1.585** | **0.816** | **42.957** | **0.849** | **0.018** | **0.184** |

Tables S7–S9 compare results across two interaction levels: Closely interactive and Naturally interactive. Our method consistently outperforms others in geometry, texture, and occluded regions. It demonstrates superior geometric fidelity (e.g., CD, NC, F-score), texture quality (PSNR, SSIM, LPIPS), and robustness under occlusions, regardless of interaction level. These results highlight the resilience and generalizability of our approach under varying interaction conditions.

Table S8: Quantitative comparison of texture depending on level of interaction.

| Interaction | Method | PSNR ↑ | SSIM ↑ | LPIPS ↓ |
|---|---|---|---|---|
| Closely | SIFU | 14.369 | 0.781 | 0.223 |
| | SiTH | 13.683 | 0.785 | 0.243 |
| | PSHuman | 11.722 | 0.747 | 0.293 |
| | **Ours** | **16.454** | **0.805** | **0.179** |
| Natural | SIFU | 15.586 | 0.799 | 0.192 |
| | SiTH | 13.851 | 0.790 | 0.228 |
| | PSHuman | 11.278 | 0.740 | 0.309 |
| | **Ours** | **16.741** | **0.818** | **0.166** |

Table S9: Quantitative comparison within occluded regions depending on level of interaction.

| Interaction | Method | Norm $L2$ ↓ | PSNR ↑ | SSIM ↑ |
|---|---|---|---|---|
| Closely | SIFU | 0.223 | 5.745 | 0.569 |
| | SiTH | 0.218 | 5.900 | 0.551 |
| | PSHuman | 0.258 | 4.344 | 0.529 |
| | DeepMultiCap | 0.219 | - | - |
| | **Ours** | **0.153** | **8.082** | **0.610** |
| Natural | SIFU | 0.184 | 6.359 | 0.554 |
| | SiTH | 0.187 | 6.557 | 0.532 |
| | PSHuman | 0.249 | 4.757 | 0.501 |
| | DeepMultiCap | 0.216 | - | - |
| | **Ours** | **0.138** | **8.358** | **0.599** |

## C.4 GENERALIZATION TO OUT-OF-DISTRIBUTION HUMANS

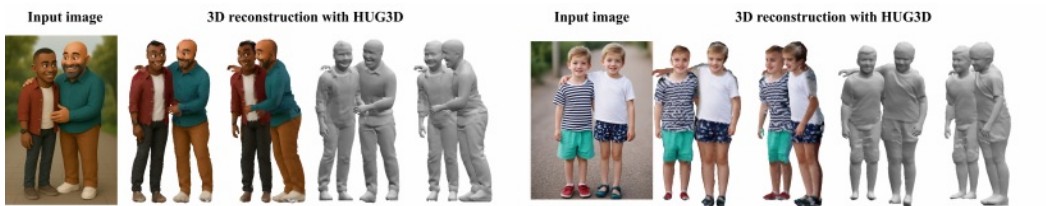

Figure S17: Qualitative results demonstrating HUG3D's generalization capability to novel human types, including stylized 3D characters and children. Despite domain differences, our method produces structurally plausible and semantically consistent outputs.

To assess the robustness of our method, we tested HUG3D on novel human inputs, including stylized 3D characters and children—categories not present during training. As shown in Fig. S23, while minor mismatches in body proportions may occur due to distribution shifts, our model still generates geometrically plausible and semantically coherent outputs. These results highlight the strong generalization ability of HUG3D, even in challenging and unseen scenarios.

## C.5 RESULTS FROM MULTIPLE VIEWS

Figs. S18 and S19 show qualitative renderings of our reconstructed textured 3D mesh from a broad set of viewpoints. We visualize both training views (with gray backgrounds) and novel views (with white backgrounds), sampled across varying camera positions: elevations of {-45°, 0°, 45°} and azimuths of {0°, 45°, 90°, 135°, 180°, 225°, 270°, 315°}. These results demonstrate the model's strong generalization capability to unseen perspectives for both normal maps and RGB images.

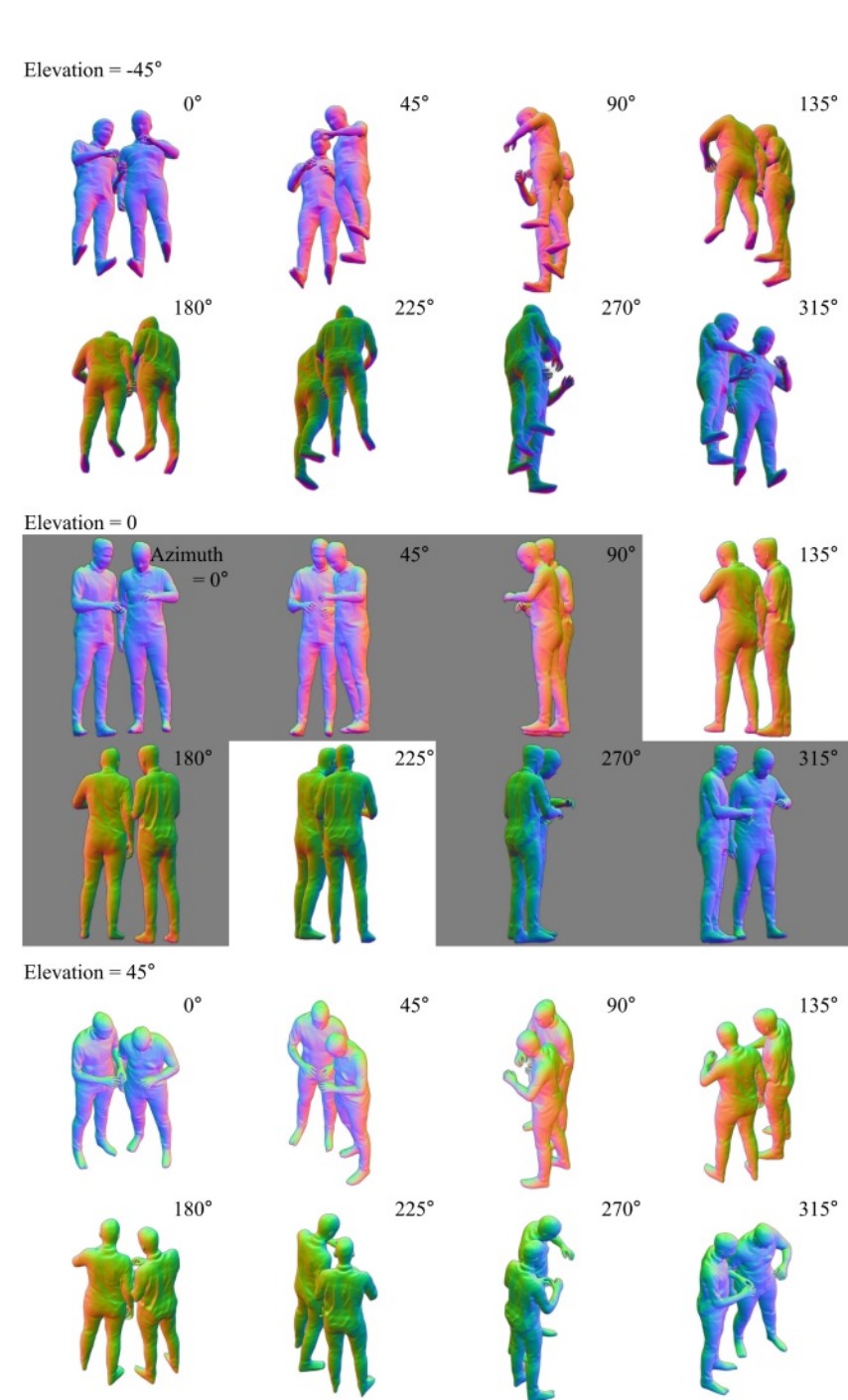

Figure S18: Normal maps rendered from multiple viewpoints of our reconstructed textured 3D mesh, including both training views (gray background) and novel views (white background).

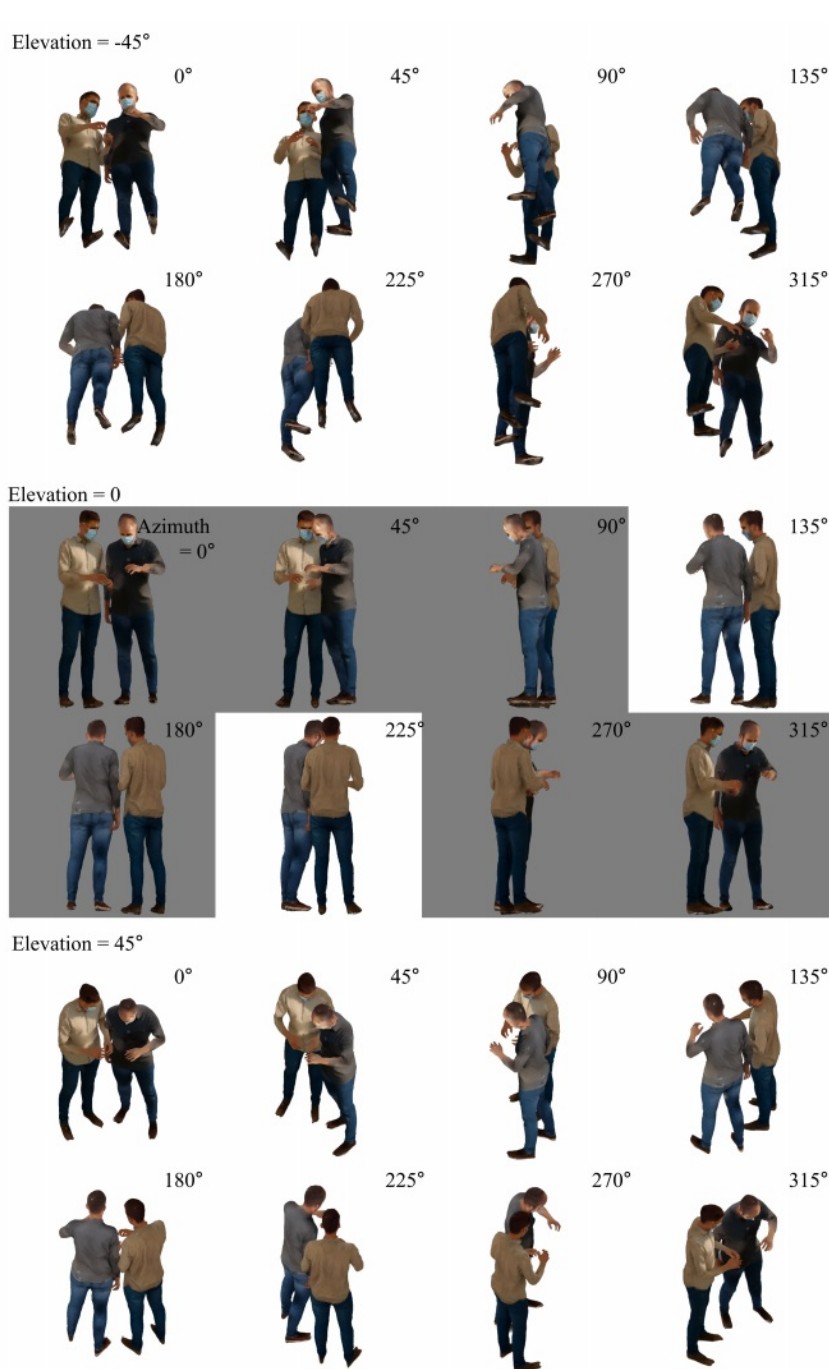

Figure S19: RGB renderings from multiple viewpoints of our reconstructed textured 3D mesh, including both training views (gray background) and novel views (white background).

## D   RESULTS FROM EACH COMPONENT

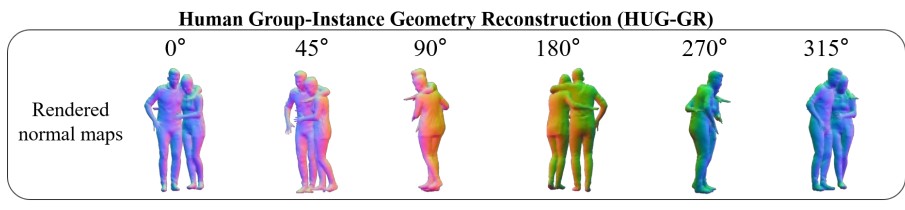
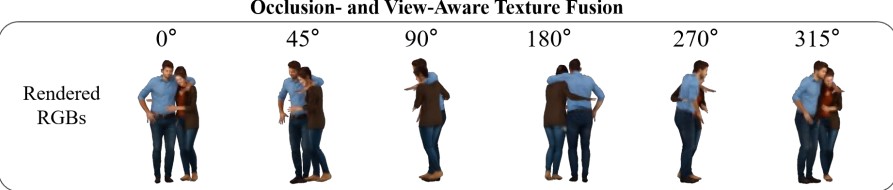

Figure S20: Example of results from each component of our HUG3D.

Fig. S20 presents qualitative outputs from each stage of our framework. (1) *SMPL-X Estimation and Instance Segmentation* produce parametric body models and segmentation masks. (2) *Canonical Perspective-to-Orthographic View Transformation (Pers2Ortho)* enables reprojection of RGB images to a shared canonical view. (3) *Human Group-Instance Multi-View Diffusion (HUG-MVD)* generates multi-view consistent RGB and normal maps. (4) *Human Group-Instance Geometry Reconstruction (HUG-GR)* reconstructs accurate 3D meshes of multiple human subjects. (5) *Occlusion- and View-Aware Texture Fusion* synthesizes high-quality textured meshes by integrating multi-view information while handling occlusions and viewpoint variations.

# E    ADDITIONAL ABLATION STUDIES AND ANALYSIS

## E.1    ROBUST SMPL-X ESTIMATION (ROBUDDI)

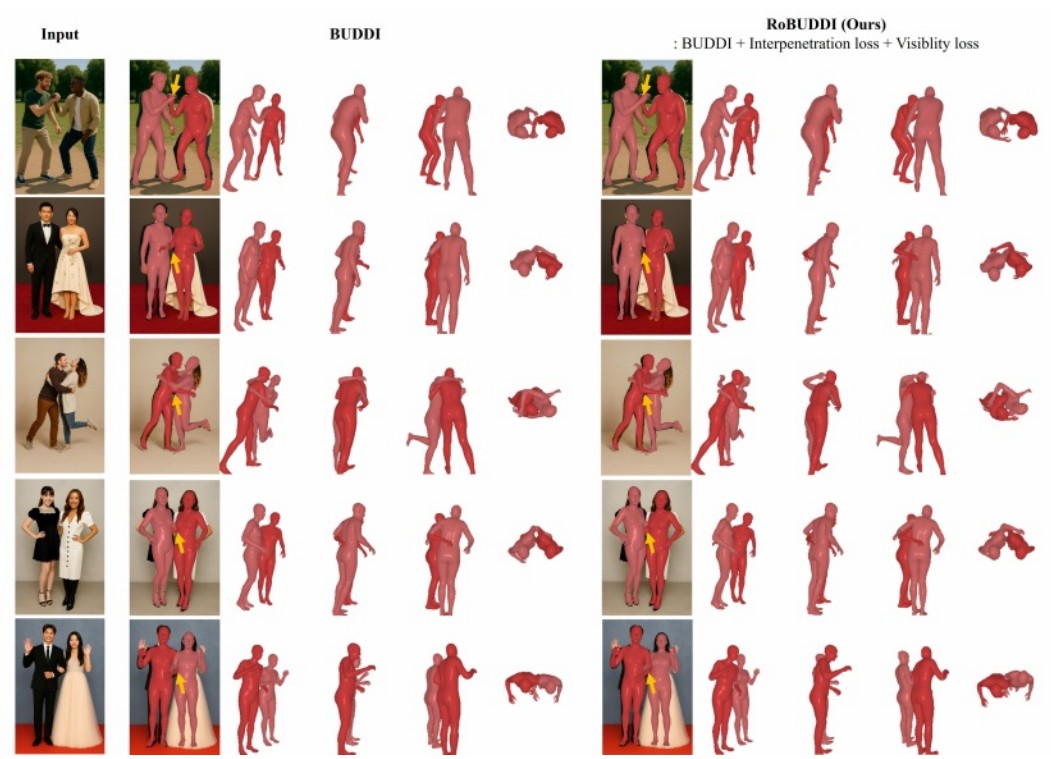

Figure S21: Qualitative comparison of our SMPL-X fitting (RoBUDDI) against BUDDI (Müller et al., 2024). BUDDI exhibits visible interpenetrations between interacting subjects (yellow arrows), whereas our RoBUDDI produces more physically plausible results.

Table S10: Quantitative comparison of SMPL-X fitting accuracy on the MultiHuman dataset (Zheng et al., 2021). Metrics include mean per-joint position error (MPJPE), its Procrustes-aligned variant (PA-MPJPE), and mean vertex error (MVE).

| Method | MPJPE ↓ | PA-MPJPE ↓ | MVE ↓ |
|---|---|---|---|
| BEV | 13.178 | 12.704 | 10.570 |
| BUDDI | 13.162 | 12.695 | 10.591 |
| **Ours (RoBUDDI)** | **13.139** | **12.673** | **10.566** |

We evaluate our proposed RoBUDDI on the MultiHuman dataset (Zheng et al., 2021) and compare it against BEV (Sun et al., 2022) and BUDDI (Müller et al., 2024). As shown in Tab. S10, RoBUDDI achieves lower MPJPE, PA-MPJPE, and MVE, demonstrating superior accuracy in 3D pose and shape estimation.

In addition to quantitative improvements, our method shows qualitative benefits as illustrated in Fig. S21. While BUDDI suffers from interpenetration artifacts between closely interacting subjects (yellow arrows), our RoBUDDI, enhanced with interpenetration and visibility-aware losses, yields more physically plausible and realistic 3D reconstructions.

## E.2    CANONICAL PERSPECTIVE-TO-ORTHOGRAPHIC VIEW TRANSFORM (PERS2ORTHO)

**Depth Edge-Aware Uncertain Point Filtering.** As shown in Fig. S7, removing uncertain points helps reduce jagged contours and ghosting artifacts near object boundaries after reprojection.

**Depth-Aware Visible Point Selection.** As shown in Fig. S8, this strategy filters out occluded or background points, retaining only those in front of the mesh surface and visible from the target camera view.

### E.3 HUMAN GROUP-INSTANCE MULTI-VIEW DIFFUSION (HUG-MVD)

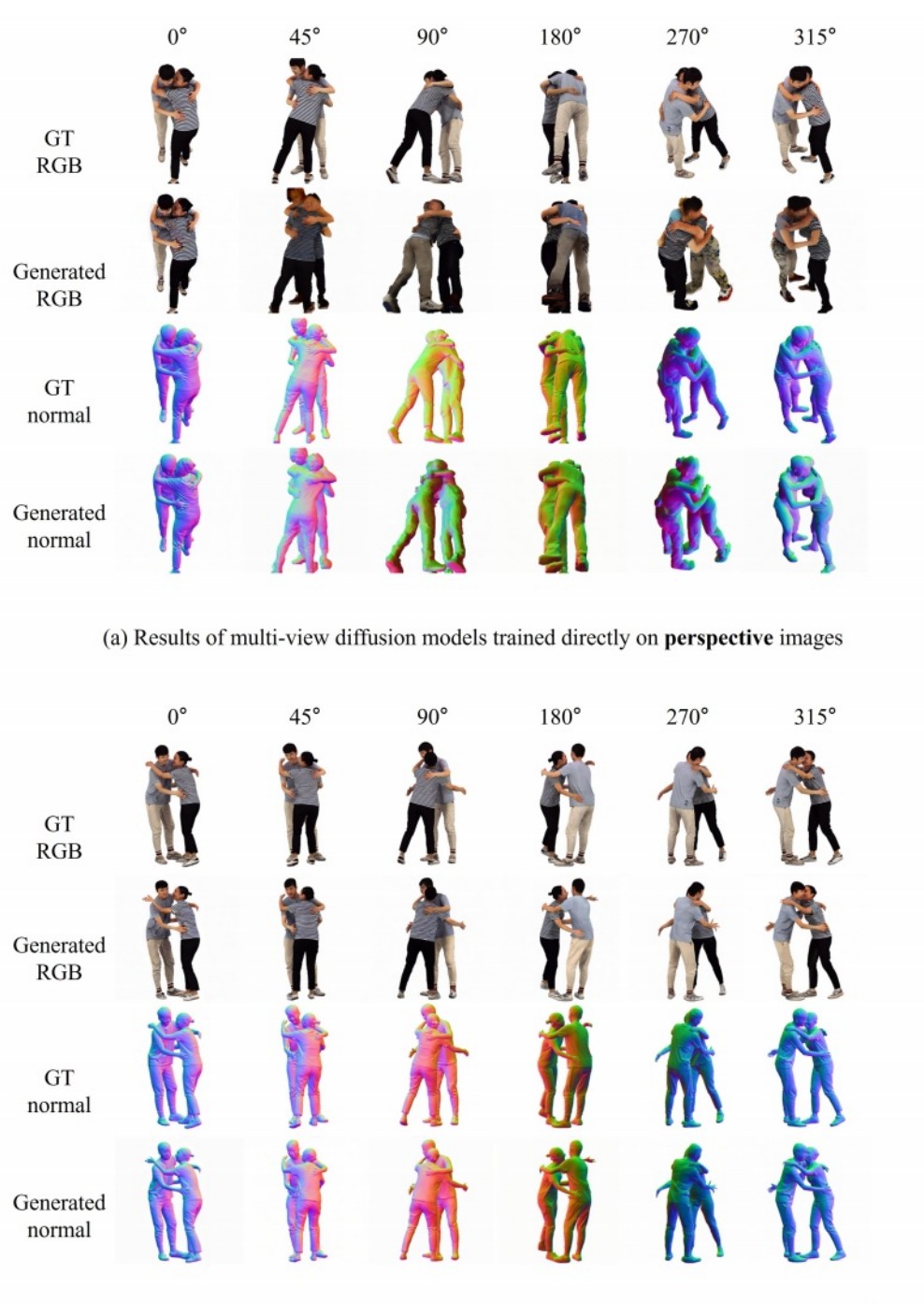

(a) Results of multi-view diffusion models trained directly on **perspective** images

(b) Results of multi-view diffusion models trained on **orthographic** images (HUG-MVD)

Figure S22: Comparison of results from multi-view diffusion models trained directly on perspective images vs. models trained on orthographic images.

**Perspective multi-view diffusion vs. Orthographic multi-view diffusion.** We compare the results of multi-view diffusion models trained directly on perspective images with those trained on orthographic images in Fig. S22 with the same training settings. Due to the limited amount of ground-truth group data, the geometric complexity of group scenes, and the constrained capacity of the base diffusion model, models trained on perspective images often fail to produce plausible outputs (Fig. S22(a)). These failures manifest as artifacts such as twisted limbs and mixed clothing textures. In contrast, our multi-view diffusion model trained on orthographic images performs significantly better under limited data settings (Fig. S22(b)). This highlights the effectiveness of our strategy, which first transforms perspective images into orthographic views and then applies multi-view diffusion in a canonical space.

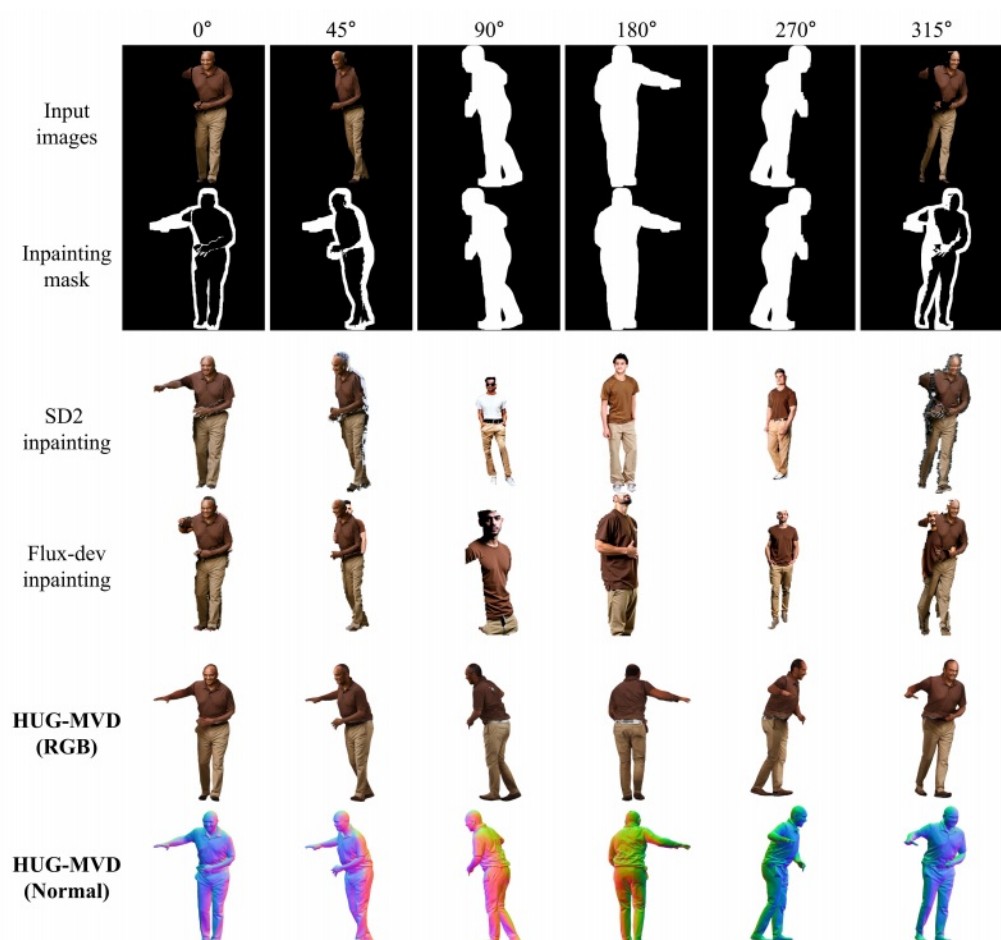

Figure S23: Comparison between state-of-the-art diffusion-based inpainting methods and our HUG-MVD framework for multi-view consistent inpainting and generation.

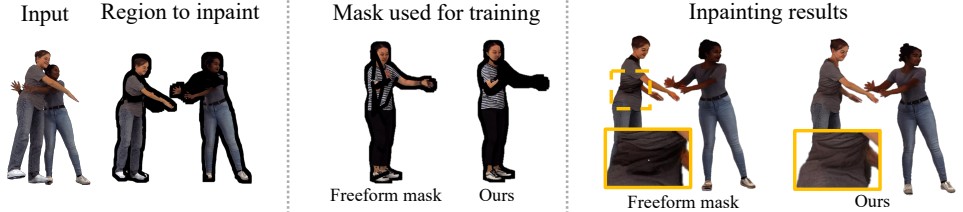

Figure S24: Ablation on mask types for training HUG-MVD. Our occlusion-simulated masks enhance inpainting compared to freeform masks.

**Comparison with state-of-the-art diffusion-based inpainting.** Fig. S23 shows a detailed comparison between current state-of-the-art diffusion-based inpainting methods (Rombach et al., 2022;

Labs, 2024) and our HUG-MVD, which achieves multi-view consistent inpainting and generation. Unlike existing inpainting approaches, HUG-MVD (1) produces multi-view consistent results, (2) maintains pose consistency without anatomical errors, and (3) additionally performs normal map inpainting—representing a significant advancement.

While several prior works address multi-view consistent inpainting, their tasks differ and are less suitable for human occlusion completion. For instance, MVInpainter (Cao et al., 2024) requires a consistent multi-view background and an inpainted object visible in the first view, and Instant3Dit (Barda et al., 2024) demands a 3D object as input. In contrast, our method only requires a single occluded human image as input, which is a more challenging and realistic scenario.

**Training MVD with occlusion-aware masks and freeform masks.** In fig. S24, we evaluate our inpainting training strategy for HUG-MVD by comparing mask generation schemes. Since freeform masks do not reflect real world occlusion patterns, it biases the model towards learning mask artifacts and producing visibly unnatural inpainted regions. In contrast, our method enables more natural, artifact-free inpainting.

### E.4 Occlusion- and View-Aware Texture Fusion

**View-Aware Face Restoration.** As shown in Fig. S13, our method effectively refines facial regions captured from extreme angles or under occlusion, such as back views, which often exhibit degraded appearances.

**Occlusion-Aware Blending.** As illustrated in Fig. S14, our method effectively prevents ghosting and bleeding artifacts near occlusion boundaries.

### E.5 Efficiency Analysis

Table S11: Comparison of inference time and memory usage across baseline methods. Our method achieves superior performance while operating within the range of existing methods.

| Metric | SIFU | ECON | SiTH | PSHuman | DeepMultiCap | MultiPly | HUG3D (Ours) |
|---|---|---|---|---|---|---|---|
| Elapsed Time (s) | 333.79 | 80.09 | 148.01 | 128.47 | 42.35 | 27907.67 | 216.32 |
| Peak VRAM (GB) | 7.31 | 5.44 | 17.79 | 32.12 | 1.37 | 5.75 | 34.76 |

Table S12: Average elapsed time and peak VRAM usage for each pipeline stage.

| Metric | Pers2Ortho | HUG-MVD | HUG-GR | Texture Fusion |
|---|---|---|---|---|
| Elapsed Time (s) | 16.20 | 60.16 | 125.46 | 14.49 |
| Peak VRAM (GB) | 14.40 | 34.76 | 7.58 | 4.95 |

We provide a comparison of inference time and memory usage across baseline methods in Tab. S11. While our end-to-end inference time of 216 seconds per image is within the range of existing methods, this represents a reasonable trade-off, as our approach substantially outperforms them in reconstruction fidelity and physical plausibility. Also, the runtime scales linearly with the number of subjects, rather than exponentially.

We also measured the elapsed time and peak VRAM usage for each stage in our proposed method as shown in Tab. S12 with NVDIA A100. We observed that the most significant bottlenecks were identified to be HUG-GR (time-wise) and HUG-MVD (peak VRAM-wise), with each stage consuming 125.46 seconds and 34.76 GB of VRAM respectively.

### E.6 Statistical Signficance Analysis

We conducted Wilcoxon signed-rank tests (Wilcoxon, 1945) to assess statistical significance across all metrics in Tabs. 1, 2, 3. As shown in Tab. S13, the p-values confirm that our method consistently outperforms all baselines with statistically significant (p value < 0.001) differences across geometry, texture, and occlusion handling metrics.

Table S13: Wilcoxon signed-rank test results (p-values) across all evaluation metrics, confirming statistically significant improvements of our method over baselines.

| Method | CD | P2S | NC | F-score | bbox-IoU | Norm $L2$ | CP | PSNR | SSIM | LPIPS | Occ.Norm $L2$ | Occ.PSNR | Occ.SSIM |
|---|---|---|---|---|---|---|---|---|---|---|---|---|---|
| SIFU | 1.8e-09 | 3.3e-08 | 3.9e-14 | 3.9e-10 | 1.6e-07 | 2.0e-39 | 1.1e-05 | 3.8e-27 | 1.7e-38 | 5.1e-37 | 5.7e-10 | 5.9e-10 | 1.9e-11 |
| SiTH | 3.6e-14 | 3.8e-14 | 3.6e-14 | 5.8e-14 | 4.6e-13 | 1.4e-51 | 8.2e-04 | 1.1e-48 | 2.2e-38 | 1.5e-51 | 1.3e-14 | 4.2e-14 | 1.4e-13 |
| PSHuman | 3.6e-14 | 3.6e-14 | 3.6e-14 | 5.2e-14 | 2.7e-13 | 1.4e-51 | 1.2e-08 | 1.4e-51 | 1.7e-51 | 1.4e-51 | 1.9e-18 | 1.4e-18 | 5.2e-17 |
| DeepMultiCap | 7.8e-13 | 1.1e-08 | 2.6e-12 | 1.3e-13 | 1.4e-13 | 1.4e-46 | 5.3e-06 | 3.5e-50 | 1.2e-27 | 5.2e-47 | 5.9e-11 | 1.1e-14 | 1.5e-14 |

# F    LICENSES FOR EXISTING ASSETS

## F.1    LIBRARIES

Table S14: Libraries used in the paper

| Library | Link to license |
|---|---|
| Pytorch (Paszke et al., 2019) | https://github.com/pytorch/pytorch/blob/main/LICENSE |
| Pytorch3D (Ravi et al., 2020) | https://github.com/facebookresearch/pytorch3d/blob/main/LICENSE |
| Diffusers (von Platen et al., 2022) | https://github.com/huggingface/diffusers/blob/main/LICENSE |

The libraries used in this work are shown in Tab. S14.

## F.2    DATASETS

Table S15: Datasets used in the paper

| Dataset | Link to license |
|---|---|
| Hi4D (Yin et al., 2023) | https://hi4d.ait.ethz.ch |
| CustomHumans (Ho et al., 2023) | https://custom-humans.ait.ethz.ch/ |
| THuman2.0 (Yu et al., 2021) | https://github.com/ytrock/THuman2.0-Dataset/blob/main/THUman2.1_Agreement.pdf |
| MultiHuman (Zheng et al., 2021) | https://github.com/y-zheng18/MultiHuman-Dataset |

The datasets used in this work are shown in Tab. S15.

## F.3    PRETRAINED MODELS

Table S16: Pretrained Models used in the paper

| Pretrained model | Link to license |
|---|---|
| Stable Diffusion 2.1 Unclip (Rombach et al., 2022) | https://huggingface.co/stabilityai/stable-diffusion-2/blob/main/LICENSE-MODEL |
| PSHuman (Li et al., 2024b) | https://github.com/pengHTYX/PSHuman/blob/main/LICENSE.txt |
| ControlNet (Zhang et al., 2023) | https://github.com/lllyasviel/ControlNet/blob/main/LICENSE |
| CodeFormer (Zhou et al., 2022) | https://github.com/sczhou/CodeFormer/blob/master/LICENSE |
| Face detector (Deng et al., 2020) | https://github.com/serengil/retinaface/blob/master/LICENSE |

The pretrained models used in this work are shown in Tab. S16.

## G  LIMITATIONS, IMPACT AND SAFEGUARDS

### G.1  LIMITATIONS

While our approach demonstrates strong performance across various scenarios, we acknowledge several aspects that offer room for future improvement.

First, our method is trained under ambient lighting assumptions with consistent illumination across multiple views. In certain challenging cases such as low-light scenes or strong lighting contrast, minor failures may occur, as shown in the top-left of Fig. 4 in the main paper, the back side of the person appears relatively dark. We note that our method is a proof of concept, and these issues can potentially be mitigated through data augmentation, more diverse training data, or incorporation of synthetic lighting variations.

Second, our method focuses on inter-human occlusion but does not yet explicitly model object-induced occlusions. In cases where a person is holding or interacting with a object, the model may fail to recognize the object as separate and instead reconstruct it as part of the body, resulting in distorted geometry (see bottom-right of Fig. 4 in the main paper). We plan to address this limitation in future work by incorporating object-aware reasoning.

Third, in cases where the model relies on predicted SMPL-X input, errors in the pose estimation can lead to discrepancies between the input image and the reconstructed mesh. The model may tend to follow the predicted SMPL-X pose, resulting in slightly misaligned geometry with the input image. Nonetheless, as shown in Tab. S10 and Fig. S21, our method remains robust under such conditions and outperforms existing baselines.

Finally, there is currently no publicly available baseline that directly matches our problem setting—multi-human reconstruction from a single image. To allow meaningful comparisons, we carefully adapted related methods across different input modalities (e.g., single-human or multi-view approaches). Although these comparisons are not perfectly aligned, they offer reasonable context. We also note that relevant recent work such as (Cha et al., 2024) was not included due to the lack of released implementation.

### G.2  IMPACT AND SAFEGUARDS

This work can have significant potential across fields such as virtual reality, gaming, telepresence, digital fashion, and medical imaging. However, the ability to generate lifelike 3D representations from minimal input raises important ethical concerns around consent, data ownership, and control over digital likenesses. Moreover, the generated normal maps or 3D mesh can be used to infer sensitive biological data of the individual. It is therefore essential to limit access to the model through controlled licensing agreements and establish guidelines centered on the consent of the input image provider to minimize these concerns.

