# OpenReview forum: "HUG3D: Human Group-Aware 3D Reconstruction from a Single Image with Physical Interaction"
_ICLR.cc/2026/Conference — ICLR 2026 Conference Withdrawn Submission_

### Official Review · Reviewer_MyME · 2025-10-28

**Soundness:** 3
**Presentation:** 2
**Contribution:** 3
**Rating:** 4
**Confidence:** 5

**Summary:**

The paper presents a multi-human reconstruction framework that first performs instant segmentation and SMPL-X fitting to obtain initial human conditions, followed by a multi-instance diffusion model to generate multi-view images for each human instance. Finally, the geometry of each person is reconstructed from these generated multi-view images.

**Strengths:**

The proposed pipeline is technically plausible, and the qualitative results demonstrate that the method can produce reasonable multi-human reconstructions. The idea of combining multi-instance diffusion with multi-view reconstruction is conceptually sound and has potential for further development.

**Weaknesses:**

The paper claims to address the reconstruction of groups of people, but all presented results involve only two individuals. If the method is limited to reconstructing pairs of people, the authors should moderate their claims accordingly and revise the title and presentation to reflect the true scope.

The proposed framework is complex and involves several sequential stages, including segmentation, SMPL-X fitting, and multi-view diffusion. Errors in intermediate stages may propagate and significantly affect the final reconstruction. The paper does not discuss how robustness is maintained against variations or inaccuracies in intermediate results.

Multi-view diffusion models are known to struggle with maintaining cross-view consistency. Since the proposed approach relies on partial observations as input, this challenge may be exacerbated. The paper should provide analysis or evidence demonstrating how view consistency is ensured.

The qualitative results in the paper are presented at very small image sizes, making it difficult to visually assess the reconstruction quality and consistency. Higher-resolution visualizations or quantitative evaluations would strengthen the paper.

**Questions:**

Please see the weakness

---

### Official Review · Reviewer_ExtP · 2025-10-31

**Soundness:** 3
**Presentation:** 3
**Contribution:** 3
**Rating:** 6
**Confidence:** 4

**Summary:**

Authors tackled a challenging problem of 3D reconstructing multi-persons from single images. The proposed pipeline has three main stages: (1) Pers2Ortho that converts the single perspective RGBs into a canonical multi-view orthographic scene, (2) HUG-MVD that predicts missing geometry and texture for multi-persons, (3) HUG-GR that optimizes meshes using physical priors. Experiments are conducted on MultiHuman dataset and the method is compared to competitive single human reconstruction baselines such as ECON, SIFU, PSHuman. HUG3D outperforms such baselines.

**Strengths:**

Novel idea: Involving Pers2Ortho and making partial 3D information as the condition for multiview diffusion seems like an interesting and sound idea which can help multi-view reasoning from a single RGB image.

Good performance: Even under a single-image setting, the method outperforms multi-viewed or video-based human reconstruction baselines (ie. DeepMultiCap, Multiply). This seems impressive.

**Weaknesses:**

The method is composed of multiple stages and it is hard to analyze what happens when some modules fail.  The paper should quantify how sensitive the final result varied. Especially, Pers2Ortho still depends on the initial SMPL-X mesh fitting. If the initial fit is bad, how does the overall performance change? There is no such analysis.

The overall pipeline looks too heavy. In appendix E.5, it is reported that the overall components take more than a few minutes.

The term “Physics-aware” looks fancy, while it only considers the interpenetration loss, which is similar to penetration or contact losses frequently observed in the related literature. This is bit disappointing.

**Questions:**

Mostly, the results are shown for two-person interaction cases. How does the method work for more than 3 persons interactions?

---

### Official Review · Reviewer_RAzp · 2025-10-31

**Soundness:** 2
**Presentation:** 2
**Contribution:** 2
**Rating:** 6
**Confidence:** 4

**Summary:**

This paper proposes HUG3D to reconstruct textured 3d human groups  from a single image. Actually, a few works have existed, but the task has not been well explored until now. This work is technically well-structured, including a per2ortho module, a group-aware MVD, and a mesh refinement stage. The experiments demonstrate consistent improvement across geometry, texture and realism metric.

**Strengths:**

1. The paper includes comprehensive experiments to evaluate the effectiveness of each design, showing significant improvement against baselines.
2. This work introduces a principal way  to handle group-level context through both diffusion and optimization stages, rather than simply extending single-person pipelines.

**Weaknesses:**

1. How to optimize grouped SMPLX with only the input view, normal and depth. I think the optimization works for single human, but not trivial for multiple individuals due to severe occlusion. The visualization looks good, but is there any insight and discussion?
2. At line 174, how to initialize multiple SMPLX to prevent overlap? Although with the following refinement, but i think the initialization is also critical.   There should be more details and include some visualizations.
3. Equation 7 is confusing.  From my understanding, this loss is underspecified. It measures only the absolute distance between point pairs without considering the inside-outside relationship. Its gradient provides no corrective directions even when two meshes overlap. Moreover, when |d| < tol, loss is tol; when |d| > tol, loss is |d|. Which will encourage the distance to tol.  I do not think it makes sense.
4. The presentation could be improved, like including a demo video for 3d visualization. And it seems like the submission only shows two-person cases, without general "multiple" human generalization performance.
5. The caption in Fig. 3 should be fixed. It's very blurry.

**Questions:**

as listed in weakness.

---

### Official Review · Reviewer_pxJK · 2025-11-01

**Soundness:** 3
**Presentation:** 3
**Contribution:** 2
**Rating:** 4
**Confidence:** 4

**Summary:**

This paper proposes HUG3D, a method to generate 3D humans from single images. At the core of the method is a Human Group-aware Multi-View Diffusion (HUGMVD) net, which generates complete multi-view normals and images from single input images. An Human Group-Aware Geometric Reconstruction (HUG-GR) module is proposed to model inter-human contact, and generate 3D meshes. Finally, a 3D reconstruction module is proposed to generate 3D textured human. The authors conducted experiments on DeepMulticap dataset to verify the effectiveness of the proposed method.

**Strengths:**

The paper is easy to follow.

Synthesizing 3D humans from single images is an interesting task with practical applications.

The method is technically sound by leveraging a multiview diffusion network and a group-instance reconstruction module to model both instance level and group level geometry for multihuman reconstruction.

**Weaknesses:**

The goal is to address multi-human reconstruction, but the experiments only involve two people, with relatively simple occlusions. It remains to be demonstrated through experiments whether HUG-MVD and HUG-GR can handle group generation for three or more humans.

MultiHuman/DeepMultiCap dataset only captured three human interactions. How about the performance on the three human data?

For evaluations, how does the SMPL/SMPLX estimation accuracy affect the performance? For multihuman scene with occlusions, the SMPLX estimation may not be accurate. This issue should be discussed, especially for multihuman reconstruction.

DeepMultiCap is the first method to reconstruct multiple humans, and qualitative comparisons against DeepMultiCap are required

Demo video is required to visualize more views of the generated geometry.

**Questions:**

The experiments are conducted on human subjects with relatively tight clothing. How to handle loose clothing since the method utilizes SMPLX, while SMPL cannot model loose clothing.

---

### Note · Authors · 2025-11-13

I have read and agree with the venue's withdrawal policy on behalf of myself and my co-authors.